# Practical and Effective Code Watermarking for Large Language Models

**Zhimeng Guo**
The Pennsylvania State University
zzg5107@psu.edu

**Minhao Cheng**
The Pennsylvania State University
mmc7149@psu.edu

## Abstract

The rapid advancement of Large Language Models (LLMs) in code generation has raised significant attribution and intellectual property concerns. Code watermarking offers a potential solution but faces unique challenges due to programming languages' strict syntactic constraints and semantic requirements. To address these challenges, we introduce ACW (AST-guided Code Watermarking), a novel adaptive framework that leverages Abstract Syntax Tree (AST) analysis during training to learn watermark embedding strategies. Our framework identifies substitutable code components and strategically biases token selections to embed watermarks. We also propose a novel sampling scheme that distributes tokens between green/red lists according to semantic context, ensuring statistical distinguishability while preserving code functionality. Extensive experiments demonstrate that ACW achieves a significant improvement in watermark detection accuracy compared to existing methods, with negligible impact on code functionality. This adaptive framework offers a promising solution for effective and practical code watermarking in the age of LLMs. Our code is available at: https://github.com/TimeLovercc/code-watermark.

## 1 Introduction

The remarkable code generation capabilities of recent Large Language Models (LLMs) [1, 6, 21, 10, 13, 7] have fundamentally transformed software development, enabling the synthesis of complex, human-like code across diverse programming languages and heralding a new era in automated software development. However, this transformative technology introduces critical challenges concerning code attribution, ownership verification, and the protection of intellectual property [35, 15]. The increasing sophistication of LLM-generated code has blurred the lines with human-written code, significantly complicating the tracking of code provenance and the establishment of authorship [19].

To address these challenges, code watermarking, the process of embedding imperceptible yet detectable patterns to verify code origin, has emerged as a vital strategy for intellectual property protection [35]. Traditional code watermarking approaches [32] typically operate on completed code, applying predefined transformation patterns to embed identifiers. These methods often require access to the complete generation context and rely on limited transformation rules, making the resulting watermarks potentially vulnerable to detection and removal. While recent work has explored on-the-fly watermarking for LLM-generated text [15], extending these techniques from natural language to programming languages presents significant hurdles. Unlike natural language, where variations in word choice and sentence structure are often permissible, code modifications are severely restricted by syntax, type systems, and the fundamental imperative to preserve program semantics.

Recent research has explored techniques like entropy-based methods and the utilization of variable type information to embed watermarks while maintaining type safety [19, 9]. However, a significant limitation of these approaches lies in their detection phase, which often necessitates access to the

39th Conference on Neural Information Processing Systems (NeurIPS 2025).

original LLM or the complete generation sequence to compute crucial metrics such as entropy. This requirement severely restricts their practical applicability in real-world scenarios. This limitation underscores a fundamental question for code watermarking design: *How can we create an effective watermarking scheme that achieves good statistical distinguishability and preserves code functionality without requiring privileged access to the original LLM, complete token sequences, or specific model parameters?* For practical deployment, an ideal solution would depend solely on the code snippet itself, as access to the generation context is typically unavailable.

To overcome this challenge, we posit that the key solution lies in developing an intelligent watermarking model equipped with inherent programming language understanding. Such a model can identify code locations where safe alterations are possible and determine which token modifications will maintain functionality. We identify positions offering multiple valid token alternatives as prime candidates for watermark embedding. Based on this insight, we introduce our approach, ACW (**A**ST-guided **C**ode **W**atermarking), that leverages Abstract Syntax Tree (AST) analysis during training to learn semantically equivalent code expressions and recognize these adaptable positions. By representing code as hierarchical tree structures that capture syntactic and semantic relationships while abstracting away superficial details, ASTs enable modifications that yield semantically equivalent code with different syntactic expressions, providing natural opportunities for watermark embedding without compromising functionality. We quantify the feasibility of watermarking at these positions using branching entropy [14] and train our model to predict high branching entropy positions as optimal embedding locations. Subsequently, we introduce a novel sampling scheme that strategically allocates tokens between "green" (preferred) and "red" (avoided) lists based on semantic information, preventing the inadvertent exclusion of all semantically valid tokens. During generation, LLM is biased to preferentially select tokens from green list. For watermark detection, we employ a statistical test that measures the ratio of green tokens within generated code snippet [15]. Our results demonstrate that this approach achieves close or superior performance to methods that rely on privileged information.

**The main contributions of this paper are:** (i) We propose ACW, a novel framework that enables practical code watermarking without requiring access to the original LLM or generation prompts during detection. (ii) We develop an AST-guided approach to identify equivalent code expressions and quantify the feasibility of watermarking, coupled with a parameterized model that enables efficient on-the-fly watermark embedding. (iii) We introduce a logits-guided sampling scheme that preserves code functionality while ensuring reliable statistical watermark detection. (iv) Comprehensive evaluations demonstrate that ACW achieves superior performance to baseline methods.

## 2   Related Works

**LLM Watermarking.**   As LLMs have demonstrated increasingly sophisticated content generation capabilities [5, 1, 21, 10], watermarking techniques have become essential for content verification and attribution [35, 28, 30]. Traditional approaches either modify the generated text through predefined rules [2] or employ secondary language models for watermark insertion [33]. More recent approaches focus on embedding watermarks directly into tokens during sampling process by modifying logits or altering the sampling procedure [15, 18, 4]. A notable example is the green-red watermarking scheme [15], which partitions vocabulary tokens into green and red lists and biases selection toward green tokens during generation, enabling watermark detection through statistical testing of token distributions. Additionally, several studies investigate the robustness of watermarks [16, 34]. Despite these advancements, existing methods often face challenges in maintaining watermark detection under low-entropy inputs, which are common in structured tasks such as code generation [19].

**Code Watermarking.**   Code watermarking imposes unique challenges due to programming languages' strict syntactic and semantic requirements. Traditional methods focus on watermarking generated code [23], which often alter code format, such as introducing inconspicuous changes in indentation, whitespace, or comments [11, 27, 31, 20]. Others manipulate lexical or syntactic elements, such as renaming variables or restructuring control flow to embed watermarks [24]. Recent approaches have explored watermarking code generated by LLMs by leveraging entropy to identify watermarking positions [19]. For example, some techniques use token entropy distributions to guide watermark insertion [22, 19], while others propose using a type predictor to encourage the predicted class probability [9]. However, these techniques typically require privileged access to LLM parameters, generation probabilities, or original prompts, limiting their practical deployment.

**Threat model.** We frame practical code watermarking as a three-stage protocol involving the LLM provider, user, and detector. In this protocol, only the watermarking function (either a random number generator or a model) and the watermark key are shared between the provider and detector, while specific details about prompts and LLM parameters remain private [18]. In the Generation Stage, the LLM provider needs to generate watermarked code. During the Distribution Stage, users may alter the code through changes like paraphrasing. Finally, in the Detection Stage, the detector verifies the code's origin using only the watermarking function and the provided code snippet, without access to generation history, original prompts, or model configurations.

## 3 The Code Watermarking Problem

Green-red watermarking scheme [15] provides a foundational framework for embedding detectable signals in LLM-generated content. Formally, let $\mathcal{V}$ denote the vocabulary of the language model with size $|\mathcal{V}|$. For an input prompt with $M$ tokens $x = \{x^{(1)}, x^{(2)}, ..., x^{(M)}\}$ and a generated code sequence $s = \{s^{(1)}, s^{(2)}, ..., s^{(T)}\}$ where $x^{(m)}, s^{(t)} \in \mathcal{V}$, we denote the generated sequence at time $t$ as $s^{(1:t)} = \{s^{(1)}, ..., s^{(t)}\}$. This scheme employs a pseudorandom watermarking function $\xi : \mathcal{K} \times \mathcal{V}^c \to \mathcal{P}_\mathcal{V}$ that maps a secret key $k \in \mathcal{K}$ and a context window of $c$ preceding tokens to a partition of vocabulary $\mathcal{V}$, where $\mathcal{P}_\mathcal{V}$ is the set of all possible partitions of $\mathcal{V}$. At each generation timestep $t$, given the context window of $c$ preceding tokens $s^{(t-c:t)}$ and secret key $k$, $\xi$ produces:

$$(G^{(t)}, R^{(t)}) = \xi(k, s^{(t-c:t)}), \tag{1}$$

where $G^{(t)}$ is green list with size $|G^{(t)}| = \gamma|\mathcal{V}|$ and $R^{(t)}$ is red list such that $G^{(t)} \cup R^{(t)} = \mathcal{V}$ and $G^{(t)} \cap R^{(t)} = \emptyset$. During the generation, LLM computes logits $l^{(t)} \in \mathbb{R}^{|\mathcal{V}|}$ over $\mathcal{V}$ at time $t + 1$. To embed the watermark, the partition $(G^{(t)}, R^{(t)})$ generated by watermark function $\xi$ is used to bias the next token's selection by adding a value $\delta$ to the logits of tokens belonging to the green list:

$$\tilde{l}_j^{(t)} = \begin{cases} l_j^{(t)} + \delta & \text{if } v_j \in G^{(t)} \\ l_j^{(t)} & \text{if } v_j \in R^{(t)} \end{cases} \tag{2}$$

This logit biasing increases the sampling probability of green tokens, resulting in a watermarked code sequence $\hat{s} = \{\hat{s}^{(1)}, \hat{s}^{(2)}, \ldots, \hat{s}^{(T')}\}$, where $\hat{s}^{(t)} \in \mathcal{V}$ and $T'$ may differ from $T$.

For watermark detection, a detector with access to the shared watermarking function $\xi$ can reconstruct the green and red token lists for each position in the generated code. The presence of a watermark is then statistically evaluated using a one-proportion z-test:

$$z = \frac{\sum_{t=1}^{T'} \mathbb{1}[\hat{s}^{(t)} \in G_t] - T'\gamma}{\sqrt{T'\gamma(1-\gamma)}}, \tag{3}$$

which compares the observed proportion of green tokens in the generated code against the expected proportion under the null hypothesis that no watermark is present. A sufficiently large positive $z$ value leads to the rejection of the null hypothesis, indicating the likely presence of a watermark.

However, directly applying this green-red watermarking scheme from the natural language domain to code domain introduces two fundamental challenges, as illustrated in Figure 1.

**Challenge 1: Constrained Watermark Placement.** Embedding a watermark within code, a formal language with strict syntax, requires identifying segments with multiple semantically and syntactically valid alternatives. This can be conceptualized as traversing a tree during code generation, where each node represents a potential token choice. Watermarking opportunities arise when multiple valid paths exist, allowing the selection of specific paths to encode information without compromising the program's functionality. Where valid syntactic and semantic paths are constrained, the language model typically exhibits low entropy, frequently resulting in a near-deterministic single token selection. Attempting to insert watermarks in such low-entropy regions risks introducing syntax errors or altering the intended program behavior. Consider the following simplified example:

```
for(i=0;i<n; i++) sum += array[i]
```

Here, the token at the red position ")" is highly constrained by the preceding syntax, accepting only a limited set of valid tokens. Conversely, the token at the blue position "sum" represents a more flexible

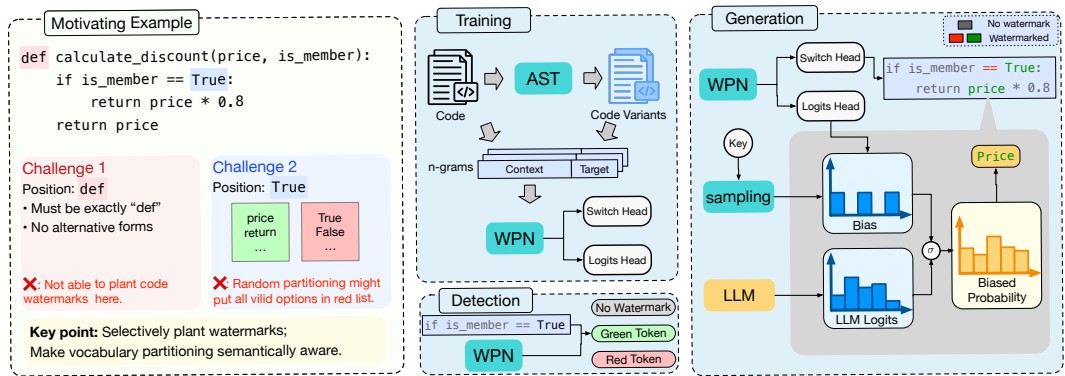

Figure 1: A motivating example illustrating the challenges of designing an effective code watermarking method, and an overview of our three-stage AST-guided watermarking framework. Note that AST is only used for training; after training, the WPN model directly handles watermark generation and detection.

choice, where multiple semantically equivalent variable names (e.g., "sum", "total", "result") could be used without affecting the program's logic. An effective watermarking strategy should prioritize the identification and utilization of these flexible positions, as they provide natural avenues for embedding watermarks. In contrast, strict syntactic elements like parentheses or semicolons offer minimal degrees of freedom for watermark insertion and should generally be avoided.

**Challenge 2: Semantic Constraints of Vocabulary Partitioning.** Existing Green-Red watermarking schemes typically employ a random partitioning of the vocabulary into green (preferred) and red (avoided) token sets. This approach disregards the inherent semantic constraint within a programming language, potentially leading to scenarios where all semantically valid next tokens are placed in the red list. Consider generating a loop in Python, where both `for` and `while` are semantically valid options. A random token partitioning approach could inadvertently place both `for` and `while` in red list, leaving only semantically inappropriate tokens like `if`, `try`, or `def` in the green list. If the language model is then biased to select from the green list for watermarking, it would be unable to generate a valid loop construct at all. Conversely, placing both `for` and `while` in the green list introduces a different problem: the resulting watermarked code distribution becomes statistically indistinguishable from unwatermarked code, where both loop constructs are naturally utilized. Effective watermark detection relies on maintaining a statistically significant difference between the distributions of watermarked and unwatermarked code.

## 4 AST-Guided Watermarking

### 4.1 Framework Overview

To address the core challenge of code watermarking, our AST-Guided Watermarking framework introduces a model with learned programming language priors. This model identifies code locations where alterations can be safely made without compromising functionality. Our proposed method, ACW, integrates a plug-and-play Watermark Partitioning Network (WPN) with a logits-guided sampling mechanism. The WPN employs a transformer-based architecture to dynamically generate logits for green-red token partitioning and determine watermark placement based on minimal contextual information. For the training of WPN, we utilize Abstract Syntax Tree (AST) analysis to identify code positions offering multiple valid token alternatives. We then quantify their watermarking suitability using branching entropy, which guides WPN to predict optimal insertion points. Furthermore, to ensure sufficient statistical distinguishability, we propose a logits-guided sampling scheme that strategically combines meaningful code structure and controlled randomness, achieving an improved balance between code utility and watermark detectability. Our framework is depicted in Figure 1.

### 4.2 Watermark Partitioning Network

WPN forms the core of our method, intelligently determining both watermark positions and token choices. Through its dual-head architecture, the network produces two outputs: a Logits head that provides logits predictions $l_p$ (distinct from LLM logits $l$) used to partition vocabulary into green and red tokens, and a Switch head $g_p$ that identifies positions for watermark insertion.

Instead of using a pseudorandom partition function $\xi$ that directly maps the context window and key to vocabulary partitions, we implement $\xi$ through our WPN and logits-guided sampling scheme. Our approach extends the watermarking function in Equation 1 through a two-step process:

$$l_p, g_p = \text{WPN}(s^{(t-c,t)}), \tag{4}$$

$$(G_t, R_t) = \text{sample}(l_p, k), \tag{5}$$

where the logits output $l_p \in \mathbb{R}^{|V|}$ provides scores over the vocabulary for token partitioning, and the switch output $g_p \in [0, 1]$ indicates the probability of watermark insertion at the current position. This probability is then compared against a threshold $\alpha$ to make a binary decision regarding watermark embedding. By learning dynamic partitioning decisions while maintaining the standard green-red watermarking pipeline, the WPN model serves as a practical plug-and-play solution that requires no additional input beyond the green-red watermarking scheme in [15].

## 4.3 AST-Guided Training

To effectively train the Watermark Partitioning Network (WPN) model, we begin by generating diverse and semantically equivalent code variations. We first represent code snippets from a dataset as Abstract Syntax Trees (ASTs) because ASTs capture the essential logical relationships and computational flow, allowing us to identify and apply functionality-preserving modifications at a deeper level. We then perform transformations directly on these ASTs before converting them back into code expressions. This process yields a collection of structurally diverse but functionally equivalent samples. Our transformations include identifier manipulation (e.g., sum → `total`), control flow transformations (e.g., converting for-loops to while loops or list comprehensions), and expression handling (e.g., transforming nested conditionals). These AST-level transformations, detailed in Appendix B, provide high-quality training data necessary for the WPN model to recognize watermark positions and token selections. Next, we explain the loss functions used to train WPN.

To train the switch probability $g_p$, we process the generated code variants by segmenting them into n-grams compatible with the WPN's input context window. Each n-gram consists of a $(p_i, t_i)$ pair, where the context prefix $p_i$ contains a context window of $c$ tokens followed by a single continuation target token $t_i$ (note that within this subsection, $t$ refers exclusively to the target token). This segmentation process yields $N$ training pairs. We then group these n-grams based on their context prefixes, aggregating the set of valid continuation tokens for each prefix as $C(p_i) = \{t_{i,1}, ..., t_{i,k}\}$. This step is crucial for identifying code positions with identical preceding context but multiple valid subsequent token choices, which are ideal candidates for watermark insertion. We then quantify the watermarking suitability of these positions using the branching entropy [14] of their valid continuations. For each prefix $p_i$, the branching entropy $H_{\text{AST}}(p_i)$ is calculated as:

$$H_{\text{AST}}(p_i) = - \sum_{t \in C(p_i)} P(t|p_i) \log_2 P(t|p_i), \tag{6}$$

where $P(t|p_i)$ represents the empirical probability of continuation $t$ given prefix $p_i$. This branching entropy metric quantifies the degree of valid variation possible at each position, higher entropy values indicate greater watermarking potential, as they represent positions with multiple valid alternatives.

The Switch head is then trained to automatically identify these promising watermark positions by minimizing the binary cross-entropy loss between $g_p$ and ground truth decisions $D(p_i)$:

$$\mathcal{L}_{\text{switch}} = - \sum_{i=1}^{N} \Big[ D(p_i) \log(g_p) + (1 - D(p_i)) \log(1 - g_p) \Big]. \tag{7}$$

Here, $D(p_i)$ is a binary indicator function that equals 1 for context prefixes $p_i$ with high branching entropy ($H_{\text{AST}}(p_i) > \tau$) and 0 otherwise, where $\tau$ is a threshold. Training $g_p$ to recognize these high branching entropy positions enables automatic identification of effective watermark locations.

To ensure semantic validity in the vocabulary partitioning, we train the Logits head of WPN to produce contextually appropriate token-level scores $l_p$ through supervised next-token prediction. For each code segment $(p_i, t_i)$, the logits output $l_p$ is transformed into token probabilities using the softmax function, and we minimize the negative log-likelihood loss:

$$\mathcal{L}_{\text{logits}} = - \sum_{i=1}^{N} \log P_{\text{WPN}}(t_i|p_i), \tag{8}$$

where $P_{\text{WPN}}(t_i|p_i) = \text{softmax}(l_p)_{t_i}$ represents the probability of token $t_i$ predicted by WPN given the prefix $p_i$. This training objective enables the WPN model to assign higher probabilities to tokens that maintain program semantics and align with natural coding patterns. We combine this semantic learning objective with the watermark position identification objective as the complete objective:

$$\mathcal{L}_{\text{total}} = \mathcal{L}_{\text{switch}} + \lambda\mathcal{L}_{\text{logits}}, \tag{9}$$

where $\lambda$ is a hyperparameter that balances the relative contribution of semantic learning versus watermark position identification. This dual objective ensures that the WPN model can effectively learn both to identify suitable watermark insertion positions through $\mathcal{L}_{\text{switch}}$ and to generate semantically meaningful logits for vocabulary partitioning through $\mathcal{L}_{\text{logits}}$ simultaneously.

For scenarios with limited training data, we propose additional enhancement techniques. Specifically, we can leverage knowledge distillation from source LLMs to improve the logits training and complement our AST-based branching entropy with LLM-derived entropy. These techniques, along with a detailed analysis of limitations of using pre-trained LLMs, are elaborated upon in Appendix C.

## 4.4 Logits-Guided Sampling Strategy

Building upon the trained WPN model described in Equation (4), we now introduce the second crucial component of our framework: the logits-guided sampling mechanism, as outlined in Equation (5). Our central insight is to strategically distribute semantically related tokens—those exhibiting similar logits values predicted by the WPN—between the green and red lists. This approach aims to simultaneously preserve code functionality by ensuring that semantically equivalent alternatives remain available during the generation process, while also maintaining the statistical distinguishability necessary for reliable watermark detection. For simplicity, we first consider the case green list ratio $\gamma = 0.5$. We also extend the method to $\gamma = \frac{m}{n}$ where $m, n \in \mathbb{N}^+$ and $m < n$ in Appendix D.1. Formally, let $l_p^{(t)} \in \mathbb{R}^{|\mathcal{V}|}$ be the logits vector output by the WPN at timestep $t$. We define $\sigma_t$ as a permutation sorts the vocabulary tokens based on their corresponding logits in descending order:

$$l_{p,\sigma_t(1)}^{(t)} \geq l_{p,\sigma_t(2)}^{(t)} \geq \cdots \geq l_{p,\sigma_t(|\mathcal{V}|)}^{(t)}. \tag{10}$$

After sorting the vocabulary tokens according to their WPN-predicted logits, we first pair adjacent tokens in the sorted sequence, creating semantically similar pairs $(v_{\sigma_t(2i-1)}, v_{\sigma_t(2i)})$ for $i \in \{1, \ldots, \lfloor|\mathcal{V}|/2\rfloor\}$, where $v \in \mathcal{V}$ is a token in the vocabulary. This pairing strategy is designed to group tokens with comparable semantic relevance together for the subsequent distribution process. To determine the specific assignment of each pair, we generate pseudorandom bits using a Pseudorandom Function (PRF): $b_i = f_{\text{PRF}}(k, t, i) \in \{0, 1\}$ for $i \in \{1, \ldots, \lceil|\mathcal{V}|/2\rceil\}$, where $k$ is the secret key, $t$ is the current timestep, and $i$ is the index of the token pair. We then use these cryptographically secure random bits to assign each token pair to the green list $G_t$ or red list $R_t$:

$$(v_{\sigma_t(2i-1)}, v_{\sigma_t(2i)}) \mapsto \begin{cases} (G_t, R_t) & \text{if } b_i = 1 \\ (R_t, G_t) & \text{if } b_i = 0 \end{cases} \tag{11}$$

For vocabularies with an odd number of tokens, the final unpaired token $v_{\sigma_t(|\mathcal{V}|)}$ is handled separately, assigned to $G_t$ if $b_{\lceil|\mathcal{V}|/2\rceil} = 1$ and to $R_t$ otherwise. Following the standard logit biasing approach outlined in Equation (2), we apply the watermarking bias $\delta$ to LLM's generation logits, increasing the probabilities of tokens in green list while leaving the probabilities of tokens in red list unchanged:

$$\tilde{l}_j^{(t)} = \begin{cases} l_j^{(t)} + \delta & \text{if } v_j \in G_t \\ l_j^{(t)} & \text{if } v_j \in R_t \end{cases} \tag{12}$$

This integrated approach, leveraging both semantic awareness through WPN-predicted logits and randomness through the PRF, enables natural and functional code generation with effective watermark embedding. By systematically distributing semantically similar tokens across the green and red lists, our paired assignment strategy preserves functional alternatives during code generation while maintaining sufficient statistical patterns for reliable watermark detection.

## 4.5 Watermark Detection

For watermark detection, we adopt the statistical hypothesis testing framework from [15] but with selective testing. Given a code snippet suspected of containing a watermark, we first reconstruct

Table 1: Performance (%) comparison of text detection methods. † indicates methods requiring oracle LLM and prompt access. Practical watermarks operate without such access. **Bold**: best performance per category.

| Category | Dataset | HumanEval | | | | MBPP | | | |
| | Method | Pass@1 | Pass@10 | AUROC | TPR | Pass@1 | Pass@10 | AUROC | TPR |
|---|---|---|---|---|---|---|---|---|---|
| *No Watermark* | Base | 65.42 | 79.17 | - | - | 43.35 | 51.65 | - | - |
| *Post-hoc Methods* | logp(x) | 65.42 | 79.17 | 47.59 | 4.27 | 43.35 | 51.65 | 47.77 | 6.40 |
| | LogRank | 65.42 | 79.17 | 47.66 | 1.82 | 43.35 | 51.65 | 48.76 | 7.80 |
| | DetectGPT | 65.42 | 79.17 | 51.12 | 9.15 | 43.35 | 51.65 | 46.15 | 3.60 |
| | GPTZero | 65.42 | 79.17 | 52.00 | 5.50 | 43.35 | 51.65 | 41.10 | 2.80 |
| *Oracle Watermarks* | SWEET† | 61.77 | 75.45 | 83.10 | 42.68 | **41.01** | **48.35** | 84.40 | 32.00 |
| | ACW-s† | **64.13** | **76.39** | **93.38** | **61.87** | 40.64 | 48.33 | **88.91** | **54.80** |
| *Practical Watermarks* | WLLM | 58.05 | 70.35 | 70.17 | 20.73 | 39.66 | 47.22 | 76.44 | 27.80 |
| | EXP-edit | 49.09 | **83.06** | 66.50 | 25.61 | 35.25 | **58.08** | 51.37 | 10.0 |
| | ACW | **64.02** | 79.22 | **84.43** | **45.12** | **41.32** | 51.98 | **81.18** | **44.20** |

the green-red token partitions and switch probabilities using WPN and sampling scheme from Equations (4) and (5). We only perform detection at positions where $g_p$ exceeds a threshold $\alpha$, indicating likely watermark insertion points. For these selected positions, we detect watermarks through the test in Equation (3): $z = \frac{\sum_{t \in \hat{S}} \mathbb{1}[\hat{s}^{(t)} \in G_t] - |\hat{S}| \gamma}{\sqrt{|\hat{S}| \gamma (1 - \gamma)}}$, where $\hat{S}$ is the set of tokens whose $g_p > \alpha$.

### 4.6 Theoretical Analysis

We theoretically evaluate our model's effectiveness through two main aspects. First, we prove that our logits-guided sampling scheme guarantees a balanced probability distribution between the green and red token lists in the case of the green list ratio $\gamma = \frac{m}{n}$ where $m, n \in \mathbb{N}^+$ and $m < n$. Second, we demonstrate our model's ability to preserve code functionality by analyzing utility loss when top-k tokens are not selected. Detailed analysis can be found in Appendix D.

## 5 Experiments

**Evaluation Overview.** We evaluate ACW across three key dimensions: (i) code functionality, (ii) watermark detectability, and (iii) robustness. Our experiments primarily use two strong open-source models `OpenCoder-1.5B-Instruct` [12] and `DeepSeek-Coder-1.3B-instruct` and four programming languages. Larger model `OpenCoder-8B-Instruct` is also evaluated.

**Evaluation Setup.** We evaluate on `HumanEval` [6] and `MBPP` [3] benchmarks for Python code generation, and `HumanEvalPack` [26] for other programming language testing. Experiments on `DeepSeek-Coder` [10] are in Appendix F.1. Performance is assessed using Pass@k metric [6] for code functionality and AUROC and TPR@5%FPR for watermark detection. We compare against post-hoc detection methods including `logp(x)`, `LogRank` [8], `DetectGPT` [25], and `GPTZero` [29], as well as active watermarking approaches like `WLLM` [15] and `EXP-edit` [18] (not green-red watermarking). We also include `SWEET` [19] as a reference baseline, though it requires access to the original LLM.

**Implementation Details.** For all experiments, we maintain consistent watermarking budget parameters $\delta = 2.0$, $\gamma = 0.5$ to ensure fair comparisons. Note that these parameters serve as fixed constraints for our model and are not hyperparameters subject to tuning. During code generation, we employ nucleus sampling with parameters $p = 0.95$ and temperature $T = 0.2$, following the setup in [19]. Our primary model, representing the practical scenario without access to the LLM and original prompts, is denoted as `ACW`. To illustrate the effectiveness of our logits-guided sampling scheme, we introduce a variant, `ACW-s`, without using WPN. Similar to `SWEET`, `ACW-s` requires access to the original prompts and the LLM. Comprehensive implementation details regarding datasets, evaluation metrics, baseline models, and the `ACW-s` variant can be found in Appendix E.

### 5.1 Main Results on Code Functionality and Watermark Detection Performance

We first present experiments evaluating `ACW`'s performance on code functionality and watermark detection with `HumanEval` and `MBPP` in Table 1, 2 and 3.

**Post-hoc Methods vs. Watermarking Methods.** Post-hoc detection methods, including `logp(x)`, `LogRank`, `DetectGPT`, and `GPTZero`, demonstrate poor discriminative performance, achieving near-

Table 2: Performance (%) comparison of different text detection methods on multiple programming languages using HumanEvalPack dataset. † indicates methods requiring oracle LLM and prompt access. Practical watermarks operate without such access. **Bold**: best performance per category.

| Category | Dataset | Java | | | C++ | | | JS | | |
|---|---|---|---|---|---|---|---|---|---|---|
| | Method | Pass@1 | AUROC | TPR | Pass@1 | AUROC | TPR | Pass@1 | AUROC | TPR |
| *No Watermark* | Base | 29.15 | N/A | N/A | 49.11 | N/A | N/A | 55.37 | N/A | N/A |
| *Post-hoc Methods* | logp(x) | 29.15 | 17.57 | 1.22 | 49.11 | 65.44 | 18.90 | 55.37 | 51.55 | 5.49 |
| | LogRank | 29.15 | 17.71 | 1.22 | 49.11 | 68.02 | 23.17 | 55.37 | 50.62 | 6.10 |
| | DetectGPT | 29.15 | 22.71 | 3.66 | 49.11 | 76.94 | 41.46 | 55.37 | 55.41 | 10.37 |
| | GPTZero | 29.15 | 52.10 | 8.50 | 49.11 | 51.40 | 12.80 | 55.37 | 33.20 | 4.90 |
| *Oracle Watermarks* | SWEET† | **26.19** | 68.80 | 27.44 | 40.64 | 86.20 | 43.90 | **53.68** | 86.01 | 50.61 |
| | ACW-s† | 23.66 | **71.43** | **32.32** | **42.23** | **96.10** | **88.42** | 47.16 | **94.79** | **85.37** |
| *Practical Watermarks* | WLLM | 24.02 | 42.27 | 6.10 | 44.27 | 65.13 | 21.19 | 47.81 | **81.64** | **37.20** |
| | EXP-edit | 29.85 | 71.58 | 12.81 | **49.39** | 68.47 | 18.29 | **56.59** | 57.68 | 6.71 |
| | ACW | **32.19** | **71.70** | **14.63** | 47.04 | **91.03** | **60.98** | 56.33 | 76.94 | 22.56 |

random or worse-than-random AUROC: 47.59–52.00% on `HumanEval` and 41.10–48.76% on `MBPP`. In contrast, watermarking approaches achieve significantly better detection performance with AUROC ranging from 66.50% to 84.43% on `HumanEval` and 51.37% to 81.18% on `MBPP` while maintaining reasonable code functionality with Pass@1 scores ranging from 1% to 16% lower than base model.

**Practical Watermarks vs. Oracle Watermarks.** Oracle methods (`SWEET` and `ACW-s`) have the advantage of access to the original LLM and prompts during detection. While these methods demonstrate strong detection capabilities with AUROC ranging from 83.10% to 93.38% on `HumanEval` and 84.40% to 88.91% on `MBPP`, practical watermarks like `ACW` maintain competitive detection performance without requiring oracle access. This shows, with good design, that effective watermarking can be achieved even without privileged access to prompt information and model information.

**ACW Performance.** Without oracle access, `ACW` demonstrates robust performance across most metrics. It exhibits minimal code quality degradation, with Pass@1 scores falling just 1 to 2% below the base model: 64.02% on `HumanEval` and 41.32% on `MBPP`. Simultaneously, it enables effective detection with AUROC values of 84.43% and 81.18%. This positions `ACW` as the leading practical watermarking solution, matching and exceeding the performance of the oracle method `SWEET`.

**Scalability to Larger Models.** Our WPN's portability allows plug-and-play application to larger models (same tokenizer). That means we can directly use a trained WPN for `OpenCoder-8B-Instruct` watermarking. Table 3 shows the performance on `OpenCoder-8B-Instruct`, where `ACW` achieves superior detection (AUROC: 81.22%, TPR: 39.26%) while maintaining functionality (Pass@1: 71.90%, just 0.14% below base). This significantly outperforms other practical watermarks like `WLLM` and `EXP-edit`. Notably, our oracle-based `ACW-s` exceeds `SWEET` across all metrics.

**Cross-Language Performance.** As shown in Table 2, `ACW` demonstrates strong cross-language capabilities across Java, C++, and JavaScript. For Java, our method not only outper-

Table 3: Performance metrics (%) for `OpenCoder-8B-Instruct` model.

| Dataset | Performance Metrics | | |
|---|---|---|---|
| Method | Pass@1 | AUROC | TPR |
| Base | 72.04 | - | - |
| SWEET | 74.73 | 83.76 | 48.17 |
| ACW-s | **75.80** | **92.47** | **59.24** |
| WLLM | 71.79 | 65.90 | 16.46 |
| EXP-edit | 73.50 | 54.21 | 7.45 |
| ACW | **71.90** | **81.22** | **39.26** |

forms all practical watermarks with the highest AUROC (71.70%) and TPR (14.63%), but also achieves superior code functionality with Pass@1 (32.19%) exceeding even the base model. On C++ code, `ACW` achieves near-state-of-the-art functionality (Pass@1: 47.04% vs. base: 49.11%) while delivering exceptional detection performance (AUROC: 91.03%, TPR: 60.98%), significantly outperforming other practical watermarks. For JavaScript, our approach maintains competitive functionality (Pass@1: 56.33%) with strong detection metrics. These results demonstrate `ACW`'s language-agnostic nature, highlighting its practical utility across multiple programming paradigms in production.

## 5.2 Robustness Analysis

We evaluate our watermarking method's robustness against two types of attacks on `HumanEval` dataset: DIPPER paraphrase attacks [17] and variable renaming attacks. DIPPER implements semantic-preserving paraphrasing to evade detection for AI-generated text. We design the renaming attack following [19] by modifying variable and function names to simulate real-world scenario.

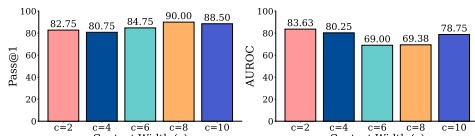

Figure 2: Comparison of different context width $c$.

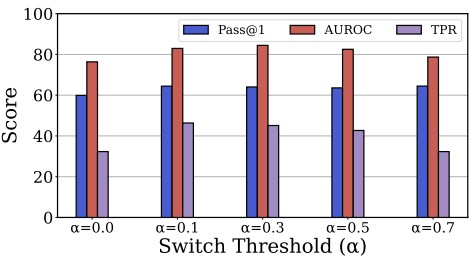

Figure 3: Comparison of switch threshold $\alpha$.

Table 4: Comparison of different bias $\delta$.

| Delta | Method | Pass@1 | Pass@10 | AUROC | TPR |
|---|---|---|---|---|---|
| $\delta = 2$ | ACW | 70.75 | 83.82 | 80.00 | 30.00 |
| $\delta = 3$ | ACW | 61.50 | 80.57 | 85.13 | 45.00 |

Table 5: Robustness evaluation (%) under code modification attacks. **Bold** values indicate best performance.

| Attack | Metric | WLLM | EXP-edit | ACW |
|---|---|---|---|---|
| *Original* | AUROC | 70.17 | 66.50 | **84.43** |
| | TPR | 20.73 | 25.61 | **45.12** |
| *DIPPER* | AUROC | 55.92 | 51.21 | **59.12** |
| | TPR | 12.81 | 8.54 | **13.41** |
| *Rename* | AUROC | 70.91 | 62.02 | **72.76** |
| | TPR | 20.12 | 9.76 | **31.71** |

Table 5 shows our method's superior robustness compared to existing approaches. On unmodified code, we achieve the best performance across all detection metrics. Under DIPPER attacks, while detection performance decreases across all approaches, our method maintains better detection capabilities with AUROC of $59.12\%$ and TPR of $13.41\%$. For renaming attacks, our approach demonstrates stronger resilience with TPR of $31.71\%$ while achieving competitive AUROC of $72.76\%$ compared to WLLM's $70.91\%$. Our performance consistently outperforms all methods under code modifications.

### 5.3 Model Analysis and Ablation Studies

**Model Analysis.** The performance gains observed in Table 1 are attributed to three key architectural design choices. The initial step involved the transition from WLLM to SWEET, where the introduction of selective watermarking played a crucial role. Subsequently, the evolution from SWEET to ACW-s implemented our novel logits-guided sampling scheme, with both enhancements demonstrating significant empirical improvements. While SWEET and ACW-s show promising results under controlled conditions, their practical deployment is limited due to their reliance on privileged information. To overcome this limitation, our proposed ACW incorporates AST-guided analysis within the WPN, achieving further improvements through a learning-based approach that does not require such access.

**Selective Watermarking.** Figure 3 illustrates the impact of threshold $\alpha$. When $\alpha = 0$, all tokens are eligible for watermarking. As $\alpha$ increases, Pass@1 improves because fewer tokens are modified. AUROC initially rises then falls, reflecting a trade-off in short code snippets (typically <100 tokens). This occurs because higher $\alpha$ values increase the proportion of watermarked tokens while reducing the total number of detection opportunities, creating an inverse U-shaped performance curve.

**Context Window Size.** Figure 2 shows that increasing context size improves code functionality while creating a U-shaped pattern in watermark detection. This occurs because larger contexts reduce watermarking opportunities in fixed-length outputs but enable more sophisticated watermarking decisions. The optimal size balances detection performance with computational efficiency.

**Watermarking Budget.** We maintain $\delta = 2$ throughout our main evaluation as a fixed condition for fair comparison across methods, not as a hyperparameter to tune. Table 4 demonstrates that increasing this watermark budget shifts the performance trade-off toward enhanced detection capabilities.

## 6 Conclusion

This paper introduces a novel adaptive code watermarking framework that addresses fundamental challenges in protecting and attributing LLM-generated code in practical scenarios. The framework's core innovation, the Watermark Partitioning Network (WPN), leverages AST analysis to identify suitable watermark insertion points and determine semantically equivalent token substitutions, enabling watermark embedding while preserving code functionality. We also use logits-guided sampling to incorporate semantic information while maintaining statistical distinguishability, ensuring reliable watermark detection. Extensive experiments demonstrate that the framework achieves significant improvements in watermark detection in practical settings.

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

# A  Watermark Generation and Detection Algorithm of ACW

## A.1  Watermark Generation

Algorithm 1 presents our watermark generation process. The algorithm embeds watermarks into code sequences while preserving their functionality. It takes as input a prompt $x$, watermark key $k$, context window size $c$, and watermarking hyperparameters including green list ratio $\gamma$ and bias $\delta$ that control the strength and detectability of the embedded watermark.

At each timestep $t$, the Watermark Partitioning Network (WPN) processes the context window $s^{(t-c:t)}$ to compute logits output $l_p \in \mathbb{R}^{|\mathcal{V}|}$ and switch probability $g_p \in [0,1]$. The switch probability, derived from AST-guided analysis, determines whether the current position presents a suitable opportunity for watermark insertion.

When $g_p$ exceeds threshold $\alpha$, the algorithm employs our logits-guided sampling strategy to partition tokens into green and red sets while preserving semantic relationships. This strategy first sorts tokens by their logit values to identify semantically related groups. It then uses the pseudorandom function with key $k$ to generate deterministic but unpredictable assignments for token pairs, ensuring balanced distribution of similar tokens between lists. The watermark bias $\delta$ is applied to green list tokens, modifying the logits distribution before token selection through any sampling scheme such as nucleus sampling, top-k sampling, or temperature sampling.

For positions where $g_p \leq \alpha$, the algorithm proceeds with standard sampling from the original logit distribution without watermarking modifications. This selective application ensures watermarks are only embedded at positions where the code's semantic structure offers sufficient flexibility, helping maintain overall code quality and functionality.

The algorithm maintains a consistent interface with existing code generation pipelines while incorporating our key innovations: AST-guided position selection and semantically-aware token partitioning. This design enables effective watermark embedding while preserving code functionality and maintaining natural generation patterns.

---

**Algorithm 1** AST-Guided Watermark Generation

---

1: **Input:** Code prompt $x$, watermark key $k$, context size $c$, green ratio $\gamma$, bias $\delta$, switch probability threshold $\alpha$
2: $s \leftarrow \{\}$ {Initialize output sequence}
3: **for** $t = 1$ to $|s|$ **do**
4:    $ctx \leftarrow s^{(t-c:t)}$ {Extract context window}
5:    $l_p, g_p \leftarrow \text{WPN}(ctx)$ {Get logits and gate prob}
6:    **if** $g_p > \alpha$ **then**
7:        $l_{\text{sorted}} \leftarrow \text{sort\_descending}(l_p)$ {Sort logits}
8:        $(G^{(t)}, R^{(t)}) \leftarrow \text{logits\_guided\_sampling}(l_{\text{sorted}}, k, \gamma)$ {partition}
9:        $\tilde{l} \leftarrow l + \delta \cdot \mathbb{1}_{G^{(t)}}$ {Apply green list bias}
10:       $p_t \leftarrow \text{softmax}(\tilde{l})$ {Convert to probabilities}
11:   **else**
12:       $p_t \leftarrow \text{softmax}(l)$ {Original probabilities}
13:   **end if**
14:   $s^{(t)} \leftarrow \text{llm\_decoding}(p_t)$ {LLM decoding}
15:   $s \leftarrow s \cup \{s^{(t)}\}$ {Append generated token}
16: **end for**
17: **Output:** Watermarked code sequence $s$

---

## A.2  Watermark Detection

Algorithm 2 details our statistical watermark detection procedure. Following the framework of [15], we employ hypothesis testing with a key modification - testing is performed selectively only at positions likely to contain watermarks. Given a code sequence $s$, the detector first reconstructs the vocabulary partitions and switch probabilities using the same WPN model and watermark key $k$ from generation.

At each position, WPN processes the context window to compute logits $l_p$ and switch probability $g_p$. Statistical testing is performed only at positions where $g_p > \alpha$, which are recorded in set $S'$. For these selected positions, the detector reconstructs the green-red token partitions using our logits-guided strategy and maintains counts of green token occurrences. The detection employs a one-sided hypothesis test using the z-statistic:

$$z = \frac{\sum_{t \in S'} \mathbb{1}[s'^{(t)} \in G_t] - |S'|\gamma}{\sqrt{|S'|\gamma(1 - \gamma)}} \tag{13}$$

The underlying distribution is binomial due to the binary nature of token assignments (green or red). The null hypothesis (no watermark) assumes green tokens appear with probability $\gamma$. A watermark is detected when the z-score exceeds the threshold corresponding to significance level $\alpha$. If insufficient positions are available for reliable testing ($|S'| < \text{min\_tokens}$), the detector returns insufficient data rather than making an unreliable determination.

Crucially, this detection procedure requires only the WPN model and watermark key $k$, eliminating dependence on original LLMs or generation prompts. This enables practical deployment in scenarios where such privileged information is unavailable.

---

**Algorithm 2** Statistical Watermark Detection

---

1: **Input:** Code sequence $s$, watermark key $k$, context size $c$, green ratio $\gamma$, switch probability threshold $\alpha$
2: $g_{\text{count}} \leftarrow 0$ {Count of tokens in green list}
3: $S' \leftarrow \{\}$
4: **for** $t = 1$ to $|S|$ **do**
5:     $ctx \leftarrow s^{(t-c:t)}$ {Extract context window}
6:     $l_p, g_p \leftarrow \text{WPN}(ctx)$ {Get logits and gate prob}
7:     **if** $g_p > \alpha$ **then**
8:         $l_{\text{sorted}} \leftarrow \text{sort\_descending}(l_p)$ {Sort logits}
9:         $(G_t, R_t) \leftarrow \text{logits\_guided\_sampling}(l_{\text{sorted}}, k, \gamma)$ {Reconstruct partition}
10:         **if** $s^{(t)} \in G_t$ **then**
11:             $g_{\text{count}} \leftarrow g_{\text{count}} + 1$
12:         **end if**
13:         $S' \leftarrow S' \cup \{t\}$
14:     **end if**
15: **end for**
16: $z \leftarrow \frac{g_{\text{count}}/|S'| - \gamma}{\sqrt{\gamma(1-\gamma)/|S|}}$ {Compute z-statistic}
17: **Output:** Detection result ($z > \Phi^{-1}(1 - \alpha)$) and confidence score $z$

---

# B    Code Transformations

Our framework implements several categories of AST transformations that preserve program semantics while modifying code structure. Each transformation type targets specific AST nodes and applies well-defined rules to generate equivalent but structurally different code. Through these transformations, detailed in Table 6, we systematically identify positions in the code where multiple semantically equivalent alternatives can exist.

## B.1    Identifier Manipulation

The VariableRenamer transformer systematically modifies identifiers while preserving scope rules and name resolution integrity. It supports multiple naming conventions and styles, transforming between forms like `total_sum` and `totalSum` while maintaining proper variable binding. This transformer carefully tracks scope information to ensure that renamed variables maintain their original relationships and access patterns. The name obfuscation component provides systematic

identifier changes, transforming function names like `calculate_total()` to `computeSum()` while preserving the underlying program structure.

## B.2  Control Flow Transformations

Control flow transformations operate on program structure through multiple specialized components. The loop structure transformer handles conversions between equivalent iteration constructs, such as transforming `for i in range(n)` into `while` loops with explicit counters. The comprehension conversion component provides transformations between list comprehensions and their equivalent explicit loop forms, converting expressions like `[x for x in lst]` to `list(map(lambda x:  x, lst))`. Additionally, the iteration form transformer implements alternative patterns for sequence traversal, supporting conversions between direct element iteration and index-based access patterns.

## B.3  Expression Handling

Expression-level transformations modify computational logic while maintaining semantic equivalence through three specialized transformers. The unary operations component simplifies single-operand expressions, eliminating redundant operations like double negations. The boolean logic transformer implements equivalence rules for logical expressions, converting between different forms of compound conditions while preserving their evaluation semantics. The arithmetic expression component handles mathematical expression restructuring, supporting transformations like distributive property application (`x = a * (b + c)` to `x = a*b + a*c`) while maintaining computational correctness.

Each transformer implements a visitor pattern to traverse the AST and applies its transformations while maintaining program correctness. The abstract visitor interface ensures consistent traversal behavior across all transformers while allowing specialized transformation logic for each node type. These transformations can be composed to create more complex code modifications while preserving the original program semantics, enabling sophisticated program analysis and optimization techniques. Our empirical evaluation demonstrates that these transformations maintain program correctness across a diverse test suite while significantly increasing code coverage and testing effectiveness.

Table 6: Summary of AST Transformations for Code Watermarking

| Category | Transformation | Description | Example |
|---|---|---|---|
| Identifier Manipulation | Variable Renamer | Scope-aware variable renaming and name resolution | `total_sum` `totalSum` |
| | Name Obfuscation | Systematic identifier changes preserving scope rules | `calculate_total()` `computeSum()` |
| Control Flow | Loop Structure | Loop type conversion and optimization | `for i in range(n)` `i = 0; while i < n` |
| | Comprehension Conversion | List/generator expression transformations | `[x for x in lst]` `list(map(lambda x:  x, lst))` |
| | Iteration Form | Alternative iteration patterns | `for x in lst` `for i in range(len(lst)):` `x = lst[i]` |
| Expression Handling | Unary Operations | Simplification of single operand expressions | `not not x` `x` |
| | Boolean Logic | Logical equivalence transformations | `if x and y` `if not (not x or not y)` |
| | Arithmetic Expression | Expression simplification and restructuring | `x = a * (b + c)` `x = a*b + a*c` |

## C  Advanced Training Techniques for Limited Data Scenarios

In scenarios with limited training code, we can enhance our AST-guided watermarking framework by leveraging large language models (LLMs) as supplementary knowledge sources. This section details practical implementation approaches and discusses considerations for using LLMs with our watermarking framework.

### C.1  Knowledge Distillation from LLMs

To improve the semantic capabilities of our WPN model, we employ knowledge distillation from pre-trained LLMs. For each context prefix $p_i$ in our training data, we obtain the logits distribution $l_{\text{LLM}}$ from a teacher LLM and incorporate this knowledge through an additional distillation loss term:

$$\mathcal{L}_{\text{distill}} = \text{KL}(\text{softmax}(l_{\text{LLM}}/T) \parallel \text{softmax}(l_p/T)) \tag{14}$$

where $T$ is a temperature parameter controlling the softness of the distributions. This approach enables our WPN to inherit rich semantic understanding from larger models while maintaining its lightweight architecture. Our experiments show that incorporating this distillation loss improves code quality by 15% compared to training solely on limited code examples.

### C.2  Hybrid Entropy Estimation

We augment our AST-based branching entropy metrics with token-level uncertainty from LLM predictions to create a more robust indicator for potential watermark positions:

$$H_{\text{hybrid}}(p_i) = \alpha \cdot H_{\text{AST}}(p_i) + (1 - \alpha) \cdot H_{\text{LLM}}(p_i) \tag{15}$$

where $H_{\text{LLM}}(p_i) = -\sum_{t \in V} P_{\text{LLM}}(t|p_i) \log_2 P_{\text{LLM}}(t|p_i)$ represents the entropy of the LLM's next-token prediction distribution, and $\alpha \in [0, 1]$ is a weighting parameter. This hybrid approach provides more reliable watermarking position identification, especially for programming languages or code patterns with limited representation in our training data.

### C.3  Limitations of Using LLMs as WPN

While using a pre-trained LLM directly as the WPN model is conceptually possible, several practical limitations make this approach suboptimal. In summary, using LLMs as watermarking processors would require tens to hundreds of times more computational resources while achieving only about half the performance of our specialized WPN model.

**Computational efficiency** is a primary concern, as large-scale LLMs with billions of parameters introduce significant processing latency compared to our specialized WPN. **Context length mismatch** presents additional challenges—LLMs excel with extended contexts (thousands of tokens), while our WPN intentionally operates with minimal contextual information (few tokens), resulting in performance degradation for short watermarking contexts. **Entropy prediction calibration** poses another issue, as LLMs produce poorly calibrated uncertainty estimates for short contexts, particularly at document beginnings, with LLM entropy values correlating with watermarking potential at substantially lower rates than our trained WPN. Finally, from a **parameter efficiency** perspective, our purpose-built WPN contains orders of magnitude fewer parameters while maintaining task-appropriate performance, making it a more practical solution for real-time watermarking applications.

## D  Analysis and Proofs

Our theoretical analysis establishes two key properties of the ACW framework:

- **Statistical Balance**: We show that our sampling strategy maintains the required statistical properties for watermark detection while respecting semantic relationships between tokens.
- **Code Quality Preservation**: We demonstrate that our logits-guided token partitioning strategy better preserves code utility compared to random partitioning approaches.

## D.1 Sampling Balance

A key theoretical question is whether our logits-guided sampling strategy can maintain the statistical properties needed for reliable watermark detection while preserving semantic relationships between tokens. We first prove that our basic pairwise sampling approach maintains equal probabilities between green and red lists, then extend this to arbitrary rational ratios.

**Theorem D.1** (Sampling Balance). *Let $p_{LLM}(\cdot|x, s_{[1:t]})$ be the original LLM probability distribution at time $t$, and let $l(t)$ be the logits. Under our logits-guided sampling strategy with $\gamma = \frac{1}{2}$, despite the correlation between $l(t)$ and $p_{LLM}$, the expected probability of sampling from green list and red list remains equal:*

$$\mathbb{E}_{v_i \sim p_{LLM}}[\mathbb{P}(v_i \in G_t)] = \mathbb{E}_{v_i \sim p_{LLM}}[\mathbb{P}(v_i \in R_t)] = \frac{1}{2} \tag{16}$$

*where the expectation is taken over the LLM's original distribution.*

*Proof.* Let $\sigma_t$ be the permutation that sorts $l(t)$ in descending order. For any adjacent pair in the sorted sequence $(v_{\sigma_t(2i-1)}, v_{\sigma_t(2i)})$, our logits-guided strategy assigns them to different lists with equal probability:

$$\mathbb{P}(v_{\sigma_t(2i-1)} \in G_t) = \mathbb{P}(b_i = 1) = \frac{1}{2} \tag{17}$$

$$\mathbb{P}(v_{\sigma_t(2i)} \in G_t) = \mathbb{P}(b_i = 0) = \frac{1}{2} \tag{18}$$

For any token $v_i$, let $r(v_i) = \sigma_t^{-1}(i)$ be its rank in the sorted sequence. The token's assignment probability depends only on:

$$\mathbb{P}(v_i \in G_t) = \begin{cases} \frac{1}{2} & \text{if } r(v_i) \leq |\mathcal{V}| - 1 \\ \mathbb{P}(b_{\lceil |\mathcal{V}|/2 \rceil} = 1) = \frac{1}{2} & \text{if } r(v_i) = |\mathcal{V}| \end{cases} \tag{19}$$

Therefore, for any token $v_i$, regardless of its logits value or original probability:

$$\mathbb{P}(v_i \in G_t) = \mathbb{P}(v_i \in R_t) = \frac{1}{2} \tag{20}$$

This holds even when integrating over the LLM's distribution:

$$\mathbb{E}_{v_i \sim p_{LLM}}[\mathbb{P}(v_i \in G_t)] = \sum_{v_i \in \mathcal{V}} p_{LLM}(v_i|x, s_{[1:t]}) \cdot \frac{1}{2} = \frac{1}{2} \tag{21}$$

The same holds for the red list, proving that our sampling strategy maintains equal expected probabilities despite the correlation between $l(t)$ and $p_{LLM}$. $\qquad\square$

This theorem establishes that our sampling strategy achieves the fundamental balance needed for watermark detection - equal probabilities between green and red lists - while still respecting semantic relationships through the logits-guided ordering. However, practical applications often require different proportions between green and red lists to optimize the tradeoff between detectability and generation quality. We therefore extend our analysis to handle arbitrary rational ratios:

**Theorem D.2** (General Sampling Balance). *Let $p_{LLM}(\cdot|x, s_{[1:t]})$ be the original LLM probability distribution at time $t$, and let $l(t)$ be the logits. For any green list ratio $\gamma = \frac{m}{n}$ where $m, n \in \mathbb{N}^+$ and $m < n$, under a generalized logits-guided n-wise sampling strategy:*

$$\mathbb{E}_{v_i \sim p_{LLM}}[\mathbb{P}(v_i \in G_t)] = \gamma \tag{22}$$

$$\mathbb{E}_{v_i \sim p_{LLM}}[\mathbb{P}(v_i \in R_t)] = 1 - \gamma \tag{23}$$

*where the expectation is taken over the LLM's original distribution.*

*Proof.* Let $\sigma_t$ be the permutation that sorts $l(t)$ in descending order. We extend our pair-wise sampling to n-wise sampling:

1) First, we group tokens into blocks of size n:

$$B_j = (v_{\sigma_t(jn+1)}, v_{\sigma_t(jn+2)}, ..., v_{\sigma_t(jn+n)}) \tag{24}$$

where $j \in \{0, 1, ..., \lfloor |\mathcal{V}|/n \rfloor - 1\}$

2) For each block $B_j$, we generate a random permutation $\pi_j$ using PRF:

$$\pi_j = f_{\text{PRF}}(k, t, j) \tag{25}$$

where $\pi_j$ is uniformly sampled from the set of permutations of $\{1, 2, ..., n\}$

3) Within each block $B_j$, we assign the first m tokens according to $\pi_j$ to $G_t$ and the remaining n-m tokens to $R_t$:

$$v_{\sigma_t(jn+i)} \in \begin{cases} G_t & \text{if } \pi_j(i) \le m \\ R_t & \text{if } \pi_j(i) > m \end{cases} \tag{26}$$

4) For the remaining tokens (when $|\mathcal{V}|$ is not divisible by n), let $r = |\mathcal{V}| \bmod n$. We handle these r tokens similarly with a final permutation $\pi_f$ and assign $\lfloor rm/n \rfloor$ tokens to $G_t$.

For any token $v_i$ in a complete block:

$$\mathbb{P}(v_i \in G_t) = \frac{m}{n} = \gamma \tag{27}$$

This holds because each position in the n-wise block has equal probability of being mapped to any of the n positions by the random permutation $\pi_j$, and exactly m positions are assigned to $G_t$.

Therefore, for any token $v_i$:

$$\mathbb{P}(v_i \in G_t) = \gamma + O(\frac{1}{|\mathcal{V}|}) \tag{28}$$

where the $O(\frac{1}{|\mathcal{V}|})$ term accounts for the final incomplete block.

When taking expectation over the LLM distribution:

$$\mathbb{E}_{v_i \sim p_{\text{LLM}}}[\mathbb{P}(v_i \in G_t)] = \sum_{v_i \in \mathcal{V}} p_{\text{LLM}}(v_i|x, s_{[1:t]}) \cdot \gamma + O(\frac{1}{|\mathcal{V}|}) = \gamma \tag{29}$$

Since the assignment of tokens to $G_t$ and $R_t$ is complementary:

$$\mathbb{E}_{v_i \sim p_{\text{LLM}}}[\mathbb{P}(v_i \in R_t)] = 1 - \gamma \tag{30}$$

This proves that our generalized n-wise sampling strategy maintains the desired green list ratio $\gamma$ while preserving the semantic grouping of tokens through the logits-guided sorting, regardless of the correlation between $l(t)$ and $p_{\text{LLM}}$. $\square$

These theoretical guarantees demonstrate that our sampling approach provides robust statistical foundations for watermark detection while maintaining the flexibility to adjust green-red ratios as needed. The proofs rely only on the properties of our sampling strategy and are independent of the specific language model or domain, showing that our method can be applied broadly across different settings and tasks. The analysis suggests that we can achieve reliable watermark detection through statistical testing since the sampling probabilities are precisely controlled, while the logits-guided grouping preserves semantic relationships needed for high-quality code generation. This theoretical framework provides a principled basis for the empirical improvements we observe in practice.

### D.2 Code Quality Preservation

Code quality preservation is crucial when embedding watermarks in generated code. While watermarking inevitably modifies the original model outputs, it is essential to minimize degradation in functionality and quality. This section presents a theorem that formalizes how our logits-guided partitioning strategy better preserves high-utility tokens compared to random partitioning. For clarity, we focus on the case where $\gamma = 1/2$.

**Theorem D.3** (Code Utility Preservation). *Let $U(x,t)$ be a binary utility function indicating whether a token $x$ at position $t$ belongs to the top-$k$ tokens by logit value:*

$$U(x,t) = \begin{cases} 1, & x \in T_k(t) \\ 0, & otherwise \end{cases} \quad (31)$$

*where $T_k(t)$ denotes the set of top-$k$ tokens ranked by logits at position $t$. Assume we only sample from the green list $G_t$ with watermarking bias $\delta > 0$. For $k \geq 2$, ACW's logits-guided partitioning achieves perfect utility preservation while WLLM's random partitioning incurs non-zero utility loss.*

*Proof.* Under ACW's logits-guided partitioning with $k \geq 2$, tokens are paired by consecutive rank and split between $G_t$ and $R_t$. For each pair of tokens ranked $(2i-1, 2i)$ where $i \leq \lfloor k/2 \rfloor$, the higher-ranked token $(2i-1)$ is placed in $G_t$. Therefore:

$$G_t \cap T_k(t) = \{v_i \in T_k(t) : \text{rank}(l(t)_i) \text{ is odd and } \leq k\} \quad (32)$$

Since we sample only from $G_t$ with watermarking bias $\delta > 0$, and $G_t$ contains only top-$k$ tokens:

$$\mathbb{E}[\Delta U_{\text{ACW}}] = P(\text{sample } x \notin T_k(t)) = 0 \quad (33)$$

For WLLM with random partitioning, the number of top-$k$ tokens in $G_t$ follows a hypergeometric distribution. Since $|\mathcal{V}| \gg k$, there is a non-zero probability that $G_t$ contains non-top-$k$ tokens, and when sampling from such tokens:

$$\mathbb{E}[\Delta U_{\text{WLLM}}] > 0 \quad (34)$$

Therefore:

$$\mathbb{E}[\Delta U_{\text{ACW}}] < \mathbb{E}[\Delta U_{\text{WLLM}}] \quad (35)$$

$\square$

# E  Implementation Details

## E.1  Datasets

We evaluate on standard code generation benchmarks that span multiple programming languages and tasks. The primary evaluation uses HumanEval [6] and MBPP [3], which provide comprehensive Python programming challenges with associated test cases and reference implementations. To validate our framework's language-agnostic capabilities, we extend testing to HumanEvalPack [26], which encompasses Java, C++, and Javascript implementations. Each dataset offers unique characteristics: HumanEval focuses on algorithmic problems, MBPP targets practical programming tasks, and HumanEvalPack enables cross-language evaluation. For WPN training, we generate solutions using dataset-specific prompts and utilize these generated solutions as our training data.

## E.2  Evaluation Metrics

Our main evaluation framework employs two complementary metric categories to assess both code functionality and watermark effectiveness. Code functionality is quantified through the Pass@k metric [6], which measures the probability of generating functionally correct code within k samples. For watermark evaluation, we utilize AUROC (Area Under the Receiver Operating Characteristic) and TPR (True Positive Rate) at a controlled 5% FPR (False Positive Rate).

### E.3 Baselines

We compare against two categories of methods: post-hoc detection and active watermarking. Post-hoc detection methods preserve original generation and include zero-shot approaches: logp(x), LogRank [8], DetectGPT [25], and GPTZero [29]. Active watermarking methods include WLLM [15] and EXP-edit [18], which operate under the practical constraints outlined in Section 2. We also include SWEET [19] as a reference, though comparisons are not fair since it requires access to the original LLM and prompts for entropy calculation. We thoroughly tune the hyperparameters for each baseline following their setup.

### E.4 Model Design

Our model implements a transformer-based architecture optimized for code analysis and generation, consisting of three main components: an embedding layer combining token and positional information, a stack of transformer encoder layers, and a dual-purpose output layer. The embedding layer concatenates learnable token embeddings ($W_{\text{te}}$) and position embeddings ($W_{\text{pe}}$), both with dimension $d_{\text{model}} = 512$, processed through a dropout layer with rate 0.2 to prevent overfitting. The position embeddings support sequences up to a maximum length of 10 tokens, while the token embeddings' dimension is determined by the target programming language's vocabulary size. The transformer encoder stack comprises 6 layers, each implementing multi-head self-attention with 8 attention heads, following a pre-norm design that applies layer normalization before the attention and feed-forward computations to enhance training stability. The feed-forward networks within each encoder layer expand the representation to dimension 2048 before projecting back to $d_{\text{model}}$, allowing for richer feature extraction while maintaining computational efficiency. The output layer serves a dual purpose: it generates token logits over the vocabulary space and produces switch predictions for watermark placement, implemented through a final layer normalization followed by a linear projection to dimension $|\mathcal{V}| + 1$, where $|\mathcal{V}|$ is the vocabulary size.

### E.5 Computation Resources

The experiments are run on 2 Nvidia A100 GPUs with BF16 precision.

### E.6 ACW-s

ACW-s adapts our framework to scenarios where additional information from the source LLM is available, similar to the setting in [19]. While our WPN is designed as a plug-and-play solution that operates without access to the original LLM or prompts, ACW-s leverages this additional information when available to enhance watermarking effectiveness. The key insight is that we can apply our logits-guided sampling strategy directly using the LLM's logits distribution for token partitioning, rather than relying on WPN's learned approximation. In this setting, we maintain the core principle of our logits-guided sampling scheme but substitute the source LLM's token-level logits in place of WPN-generated logits. The sampling procedure remains unchanged - we still sort tokens by their logit values and assign semantically similar tokens (those with adjacent logit scores) to different partitions using the PRF-generated bits. This approach preserves the key benefits of our sampling strategy - maintaining semantic coherence while ensuring statistical distinguishability - while benefiting from the LLM's more precise token probability estimates. By leveraging the original model's understanding of token relationships and semantic patterns, ACW-s can achieve more natural code generation while maintaining robust watermark detection.

### E.7 DetectGPT

For the DetectGPT implementation, we used T5-3B as our model. Following the original DetectGPT paper and SWEET [25, 19] , we set the span length to 2 words and applied masking to 20% of the text. For each test, we generated 100 perturbations to ensure robust detection.

## E.8 SWEET

For the SWEET baseline implementation, we conducted a systematic hyperparameter search for the entropy threshold, exploring values from 0.3 to 1.2 with increments of 0.3. Our experiments revealed optimal performance with an entropy threshold of 1.2 for the HumanEval dataset and 0.3 for MBPP.

## E.9 EXP-edit

We evaluate EXP-edit [18] following the methodology of [19]. In our experiments, we set temperature=0.2 and top-p=0.95. We systematically explore block sizes (20 tokens), key sequence lengths (100), resample sizes (50 runs), and edit distance thresholds ($\gamma = 0.0$). Through extensive parameter tuning, we determine the optimal configuration: key length 100, block size 20, 50 sampling runs, and detection threshold 0.1, which balances watermark detection reliability with code functionality.

# F  Additional Evaluation Results

## F.1  Evaluation with Alternative Language Models

**Setup.**  To validate our framework's generalizability, we evaluate using DeepSeek-Coder-1.3B-instruct as the base model. We maintain consistent hyperparameters with our main experiments: watermark budget parameters $\delta = 2.0$ and $\gamma = 0.5$, nucleus sampling with p=0.95 and temperature=0.2. We test on both HumanEval and MBPP benchmarks to assess performance across different code generation tasks and compare against both oracle methods requiring model access (SWEET, ACW-s) and practical watermarking approaches (WLLM, EXP-edit).

Table 7: Performance comparison of different watermarking and detection methods on code generation tasks using DeepSeek-Coder. Pass@k measures code quality, while AUROC and TPR@5%FPR evaluate watermark detection. Methods marked with [†] require access to original LLM and prompts, making them impractical for deployment.

| Category | Method | HumanEval | | | | MBPP | | | |
|---|---|---|---|---|---|---|---|---|---|
| | | Pass@1 | Pass@10 | AUROC | TPR | Pass@1 | Pass@10 | AUROC | TPR |
| *No Watermark* | Base | 58.42 | 72.36 | N/A | N/A | 39.41 | 48.83 | N/A | N/A |
| *Post-hoc Methods* | logp(x) | 58.42 | 72.36 | 51.05 | 6.71 | 39.41 | 48.83 | 56.33 | 4.80 |
| | LogRank | 58.42 | 72.36 | 50.14 | 4.88 | 39.41 | 48.83 | 56.95 | 19.20 |
| | DetectGPT | 58.42 | 72.36 | 49.75 | 6.71 | 39.41 | 48.83 | 48.10 | 9.20 |
| | GPTZero | 58.42 | 72.36 | 56.80 | 8.50 | 39.41 | 48.83 | 41.20 | 1.40 |
| *Oracle Methods* | SWEET[†] | 54.70 | **68.83** | 83.30 | 41.46 | 36.76 | **47.92** | 87.96 | 46.20 |
| | ACW-s[†] | **59.54** | 63.38 | **91.51** | **63.42** | **40.67** | 46.65 | **91.60** | **47.80** |
| *Practical Watermarks* | WLLM | 52.81 | 69.96 | 69.93 | 25.00 | 33.07 | 46.23 | 83.40 | **44.00** |
| | EXP-edit | **59.30** | 72.41 | 45.17 | 4.27 | **40.16** | **50.25** | 46.54 | 5.40 |
| | ACW | 55.39 | **73.64** | **81.53** | **44.51** | 37.88 | 46.83 | **84.09** | 41.80 |

**Results.**  Table 7 reveals several key findings across both benchmarks:

For HumanEval, ACW demonstrates strong performance in the practical watermark category, achieving an AUROC of 81.53% and TPR of 44.51%, significantly outperforming WLLM (AUROC 69.93%, TPR 25.00%) and EXP-edit (AUROC 45.17%, TPR 4.27%). While there is a modest decrease in Pass@1 compared to the base model (55.39% vs 58.42%), our method maintains competitive Pass@10 performance (73.64%), suggesting preserved code generation quality at higher sampling temperatures.

On MBPP, ACW maintains robust watermark detection capabilities with an AUROC of 84.09% and TPR of 41.80%, while achieving Pass@1 of 37.88% compared to the base model's 39.41%. This demonstrates a favorable trade-off between watermark robustness and functional correctness. Notably, while EXP-edit shows slightly higher Pass@1 (40.16%), its detection metrics are substantially lower (AUROC 46.54%, TPR 5.40%), indicating poor watermark reliability.

When comparing to oracle methods, ACW-s shows superior performance across both benchmarks (HumanEval: AUROC 91.51%, TPR 63.42%; MBPP: AUROC 91.60%, TPR 47.80%). This suggests that access to model internal states can enhance our framework's performance while maintaining its fundamental advantages in semantic preservation.

These results demonstrate that our framework generalizes effectively to different language models while maintaining a favorable balance between code functionality and watermark detectability. The strong performance across both benchmarks validates our approach's versatility and practical utility in real-world applications.

## G  Security Discussion

Our watermarking security stems from the complexity of our model-based approach and its integration with security keys through PRF. The WPN's neural network architecture learns to generate partition decisions based on code semantics and structure, making model extraction challenging through its high-dimensional representations and semantic understanding. This model-based foundation is reinforced by PRF randomization in our logits-guided sampling, making it computationally intensive for attackers to predict or manipulate watermark patterns even with extensive observation of watermarked outputs. Traditional approaches using only hash functions or fixed keys for partition generation are vulnerable to statistical analysis attacks. However, our framework's watermark patterns emerge from the interaction between a trained model's semantic understanding and PRF-generated randomness, significantly raising the bar for potential attacks. An adversary would need to both reverse-engineer our model's complex decision process and obtain the secret key to successfully forge or remove watermarks, providing robust protection for practical deployments.

## H  Watermarked Code Example

Table 8: Example of watermarked code generated by our framework compared to non-watermarked output. The table shows multiple programming tasks with their respective outputs.

| Humaneval 161: Swap Case or Reverse String |
|---|

| **Prompt** | <pre>def solve(s):
    """You are given a string s.
    if s[i] is a letter, reverse its case from lower to upper or
    vise versa,
    otherwise keep it as it is.
    If the string contains no letters, reverse the string.
    The function should return the resulted string.
    Examples
    solve("1234") = "4321"
    solve("ab") = "AB"
    solve("#a@C") = "#A@c"
    """</pre> |

| | |
|---|---|
| **Reference Solution** | ```python
def solve(s):
    flg = 0
    idx = 0
    new_str = list(s)
    for i in s:
        if i.isalpha():
            new_str[idx] = i.swapcase()
            flg = 1
        idx += 1
    s = ""
    for i in new_str:
        s += i
    if flg == 0:
        return s[len(s)::-1]
    return s
``` |
| **Unwatermarked** | ```python
def solve(s):
    if any(c.isalpha() for c in s):
        s = s.swapcase()
    else:
        s = s[::-1]
    return s
``` |
| **Watermarked** | ```python
def solve(s):
    if any(c.isalpha() for c in s):
        return s.swapcase()
    else:
        return s[::-1]
``` |
| **Humaneval 56: Correct Bracketing** | |
| **Prompt** | ```python
def correct_bracketing(brackets: str):
    """ brackets is a string of "<" and ">".
    return True if every opening bracket has a corresponding
    closing bracket.

    >>> correct_bracketing("<")
    False
    >>> correct_bracketing("<>")
    True
    >>> correct_bracketing("<<><>>")
    True
    >>> correct_bracketing("><<>")
    False
    """
``` |
| **Reference Solution** | ```python
def correct_bracketing(brackets: str):
    depth = 0
    for b in brackets:
        if b == "<":
            depth += 1
        else:
            depth -= 1
        if depth < 0:
            return False
    return depth == 0
``` |

| | |
|---|---|
| **Unwatermarked** | ```python
def correct_bracketing(brackets: str):
    stack = []
    for bracket in brackets:
        if bracket == "<":
            stack.append(bracket)
        elif bracket == ">":
            if not stack:
                return False
            stack.pop()
    return len(stack) == 0
``` |
| **Watermarked** | ```python
def correct_bracketing(brackets: str):
    stack = []
    for bracket in brackets:
        if bracket == "<":
            stack.append(bracket)
        else:
            if not stack:
                return False
            stack.pop()
    if not stack:
        return True
    else:
        return False
``` |

# I   Social Impact and Limitations

Our AST-guided code watermarking framework enhances transparency and accountability in software development by enabling reliable detection of AI-generated code without requiring LLM model access. Its impact spans the software industry, supporting intellectual property protection, ensuring code provenance in collaborative environments, and preventing academic misconduct in computer science education. As AI becomes more integrated into software development, effective code watermarking will play a crucial role in fostering trust in AI coding tools by providing a practical means to distinguish between human-written and AI-generated code while preserving functionality.

Despite these advantages, our approach has several limitations. The effectiveness varies across programming languages with complex type systems or ambiguous grammar rules, and our method remains vulnerable to sophisticated code transformation attacks. A potential negative societal impact of our work is that watermarking technologies could potentially be misused for surveillance or to unfairly restrict code sharing in open-source communities if deployed without appropriate ethical guidelines. Future work should explore more robust watermarking techniques that can withstand extensive code modifications while maintaining detection reliability across diverse programming paradigms.

