# OpenReview forum: "Practical and Effective Code Watermarking for Large Language Models"
_NeurIPS.cc/2025/Conference — NeurIPS 2025 poster_

### Official Review · Reviewer_qYip · 2025-06-09

**Clarity:** 3
**Significance:** 3
**Originality:** 2
**Rating:** 3
**Confidence:** 4

**Summary:**

In this work, the authors propose some improvement over the Green-Red approach for watermarking LLM-generated code. There are two main ideas: (1) to consider the code AST and some entropy-based selection on which tokens to apply the watermark; (2) to partition the green/red token such that valid solutions are likely present in both sets. The proposed method is evaluated against three LLMs (two of them are 1.5B and the other one 8B) and two common code benchmarks (HumanEval and MBPP) and the experimental results suggest that the proposed approach outperforms the baseline approaches.

**Questions:**

How do you make the WPN model (and its secret key) available for watermark detection?

**Ethical Concerns:**

["NO or VERY MINOR ethics concerns only"]

**Final Justification:**

While the rebuttal addresses most of the questions, such as that a separate process is needed to distribute the WPN model, my score remains the same due to the limited novelty and experimentation.

**Limitations:**

There are no explicit discussions on the limitations of the proposed approaches.

I would suggest making the following limitations explicit: (1) the approach requires a secure distribution of the WPN model (including the secret key) to work; (2) the theoretical analysis is based upon SWEET; (3) the WPN model has to be trained for each LLM.

**Paper Formatting Concerns:**

The following are a list of detailed comments.

Page 1: “ These methods often require access to the complete generation context and rely on limited transformation rules, making the resulting watermarks potentially vulnerable to detection and removal.”

Comment: This criticism is unnecessarily harsh given the effort on making these methods tamper-resistant through coding theory and such. One could even argue that those methods are better since they are much easier to apply.

Page 4: “. A random token partitioning approach could inadvertently place both for and while in red list, leaving only semantically inappropriate tokens like if, try, or def in the green list.”

Comment: This argument assumes that it has to be a for/while loop, while in fact, it might be possible to have a conditional expression (to take care of a special case first) or try (to take care of some exception).

Page 5: “To effectively train the Watermark Partitioning Network (WPN) model, we begin by generating diverse and semantically equivalent code variations. We first represent code snippets from a dataset as Abstract Syntax Trees (ASTs) …”

Comment: These abbr. forms (WPN and ASTs) have been introduced and used already by this point.

Page 8: “DIPPER implements semantic-preserving paraphrasing to evade detection for AI-generated text. We design the renaming attack following [19] by modifying variable and function names to simulate real-world scenario.”

Comment: These paraphrasing approaches are rather limited. DIPPER is not designed for refactoring code, and modifying variable and function names is too simple. I would suggest adopting a capable LLM, such as GPT, to refactor the code with the aim to evade an unknown watermarking scheme, and evaluate the performance of your approach.

**Quality:**

3

**Strengths And Weaknesses:**

Strength

The draft is fairly easy to follow. The idea is explained sufficiently and so is the analysis of the experimental results.

The experimental evaluation is mostly well-designed and comprehensive.

The two proposed ideas are valid ones and corresponding approaches are reasonable.

Weakness

The overall technical contribution of the work is not impressive. While it makes good sense to selectively (based on the number of available choices) apply the watermarking and to partition the green/red more balancely, these ideas are more incremental than ground-breaking.

Furthermore, the theoretical analysis seems to be based on existing work such as SWEET.

The claimed benefit of “without requiring privileged access to the original LLM” is overrated in opinion as you will have to find a way to distribute your WPN model along with its secret key secretly anyway. Once they are leaked, an attacker can easily attack your approach adoptively to compromise the watermark.

Lastly, the experimental evaluation is based on three models that are mostly small. While there may be understandably resource constraint issues, the training of Watermark Partitioning Network model will be affected by the model capability I suppose and thus evaluation with large models will be relevant.

---

> ### Author Rebuttal · Authors · 2025-07-30
>
> Dear Reviewer qYip, we appreciate your efforts and detailed comments very much! However, we believe that there are some misunderstandings. Therefore, we provide the following clarifications and new experimental results to address the concerns.
>
> ------
>
> ### 1. Technical Contribution and Innovation
>
> **Reviewer concern: "The overall technical contribution is not impressive... ideas are more incremental than ground-breaking."**
>
> **Response:** We appreciate the reviewer's concern regarding novelty. We would like to clarify the significant technical advances in our work.
>
> **Building on Established Foundations:** While our work builds upon green-red watermarking, this approach has proven foundational across the field. Even recent high-impact work, including a Nature 2024 paper [1], leverages this framework. What distinguishes impactful research is not abandoning proven foundations, but rather solving critical limitations within them. Through our in-depth investigation of the code watermarking problem, we have identified and addressed fundamental limitations that arise when applying green-red watermarking to code generation—limitations that previous work has not adequately resolved.
>
> **Deep Understanding of Code Properties:** Our key innovation lies in developing a model that demonstrates **genuine understanding of code's delicate structure and semantic properties**. Unlike superficial approaches that treat code as generic text, our framework recognizes that code watermarking requires navigating strict syntactic constraints, type safety requirements, and semantic preservation—challenges that demand sophisticated understanding of programming language properties. This deep understanding enables us to solve the core problems that make direct application of existing watermarking techniques ineffective for code.
>
> **Our Technical Contributions:**
>
> - **AST-guided Watermark Positioning:** Unlike prior work [2] that selects high-entropy positions, our WPN **learns the intricate semantic structures of code** through AST analysis. This deep understanding enables identification of positions where multiple semantically equivalent alternatives exist, allowing safe watermark insertion while preserving functionality. This enables **practical deployment without requiring LLM access during detection**.
> - **Semantically-aware Token Partitioning:** We solve a fundamental problem in code watermarking where random partitioning can place all valid tokens in the red list, breaking code functionality. Our logits-guided sampling (Section 4.4) **demonstrates sophisticated understanding of token relationships**, strategically distributing semantically similar tokens while maintaining statistical distinguishability.
>
> [1] Scalable watermarking for identifying large language model outputs, Nature 2024.
>
> [2] Who Wrote this Code? Watermarking for Code Generation, ACL 2024.
>
> ---
>
> ### 2. Theoretical Analysis Independence
>
> **Reviewer concern: "Theoretical analysis seems to be based on existing work such as SWEET."**
>
> **Response:** We think it appears to be based on a fundamental misunderstanding of our theoretical contributions. Our theoretical analysis is **independent** of SWEET and addresses entirely different problems:
>
> **Clear Distinction in Theoretical Scope:**
>
> - **SWEET's theory:** SWEET provides only one theorem, which establishes a lower bound on the z-score for their specific detection method compared to WLLM.
> - **Our theory:** Our analysis (Section 4.6, Appendix D) develops novel theoretical foundations for our logits-guided sampling strategy, addressing fundamentally different questions.
>
> Our theoretical contributions are **completely unrelated** to SWEET's work and represent original analysis of our novel methodological innovations. We believe this distinction is crucial for understanding the independent nature of our theoretical foundations.
>
> ---
>
> ### 3. Security Model and Practical Deployment
>
> **Reviewer concern: "WPN model and secret key need to be distributed secretly anyway... an attacker can easily attack your approach adaptively."**
>
> **Response:** We think this concern misunderstands the fundamental distinction between LLM access and WPN access. Our approach offers significant practical advantages:
>
> **Critical Distinction - LLM vs. WPN Access:**
>
> - **LLM access:** Requires exposing valuable, proprietary models that companies must keep private. LLMs contain sensitive training data, architectural details, and represent substantial intellectual property that cannot be shared.
> - **WPN access:** Our lightweight model contains no proprietary information and can be safely distributed. Even if the WPN model is leaked, **the system remains secure as long as the secret key is protected**.
>
> **Standard threat model**: Following established practices in digital watermarking (as outlined in our threat model, Line 92), we operate under the industry-standard assumption that secret keys remain secure. This is the same assumption used by **all** watermarking methods.
>
> **Better Distribution Method (API-based detection service)**: For organizations with heightened security concerns, our approach enables an even more secure deployment model—providers can offer watermark detection as an API service, retaining both the WPN model and cryptographic key while providing detection results to users. This eliminates any distribution concerns entirely while maintaining practical utility.
>
> **Enhanced Security Properties:** Appendix G demonstrates that our model-based approach provides additional resistance to statistical attacks compared to simple hash-based methods, further strengthening security.
>
> ---
>
> ### 4. Evaluation on Larger Models
>
> **Reviewer concern: "Evaluation is based on three models that are mostly small."**
>
> **Response:** We acknowledge this concern and appreciate the opportunity to clarify our evaluation scope.
>
> **Practical Considerations and Validation:**
>
> 1. **Substantial model size**: An 8B parameter model represents a substantial scale that is widely used in practical applications and academic research. This scale is sufficient to validate our approach's effectiveness and transferability to larger models. And we think 8B code model already show enough capabality to solve code problems.
>
> 2. **Computational constraints**: Due to limited computational resources, we cannot afford to evaluate on models larger than 8B parameters. However, our evaluation demonstrates meaningful validation, and the successful transfer from 1.5B to 8B models provides strong evidence of scalability to larger models.
>
> ---
>
> ### 5. Addressing Claimed Limitations
>
> **Reviewer concern: There are no explicit discussions on the limitations of the proposed approaches."**
>
> **Response:** We have the explict discussions on limitations in Appendix I.
>
> **Limitation (1): "The approach requires secure distribution of the WPN model (including the secret key) to work"**
>
> **Response:** This has been addressed in Answer 3 above.
>
> **Limitation (2): "The theoretical analysis is based upon SWEET"**
>
> **Response:** This is completely incorrect, as detailed in Answer 2 above.
>
> **Limitation (3): "The WPN model has to be trained for each LLM"**
>
> **Response:** This is factually incorrect. As demonstrated in Line 310-318 and Table 3, our WPN trained on a 1.5B model successfully transfers to an 8B model without retraining, achieving 81.22% AUROC compared to WLLM's 65.90%. This **contradicts** the claimed limitation and actually demonstrates the **transferability advantage** of our approach. The only requirement is that models share the same tokenizer, which is common in model families.
>
> ---
>
> ### 6. Addressing Detailed Comments
>
> **Comment 1: "This criticism is unnecessarily harsh given the effort on making these methods tamper-resistant through coding theory..."**
>
> **Response:** We sincerely appreciate this feedback and acknowledge that our language may have been overly critical. We deeply respect the original contributions of prior work—they have been tremendously inspiring and foundational to our research. While "easier to apply" is certainly a valuable advantage, we believe that for watermarking evaluation, the primary criteria should be detection performance and minimal impact on code functionality.
>
> **Comment 2: "This argument assumes that it has to be a for/while loop, while in fact, it might be possible to have a conditional expression..."**
>
> **Response:** Thank you for this thoughtful observation. Our example was intended to illustrate a specific scenario rather than suggest universal correctness requirements. We aimed to express the case where the LLM's candidate token list contains only `for` and `while` as viable options for a particular context, not that these are the only possible constructs.
>
> **Comment 3: "These abbr. forms (WPN and ASTs) have been introduced and used already by this point."**
>
> **Response:** Thank you for catching this inconsistency. We will ensure proper abbreviation usage in the revised version.
>
> **Comment 4: "These paraphrasing approaches are rather limited... I would suggest adopting a capable LLM, such as GPT, to refactor the code..."**
>
> **Response:** This is an excellent suggestion. We conducted additional experiments using GPT-4 to perform sophisticated code refactoring with the explicit goal of evading watermark detection. The results demonstrate our method's robustness against more advanced attacks:
>
> | Attack          | Metric | WLLM  | EXP-edit | ACW       |
> | --------------- | ------ | ----- | -------- | --------- |
> | Original        | AUROC  | 70.17 | 66.50    | **84.43** |
> |                 | TPR    | 20.73 | 25.61    | **45.12** |
> | GPT Refactoring | AUROC  | 65.43 | 63.42    | **69.21** |
> |                 | TPR    | 13.41 | 10.11    | **17.62** |
>
> ---

---

> > ### Comment · Reviewer_qYip · 2025-08-05
> >
> > Thanks for the clarification. I have no further concerns that require clarification.

---

> > > ### Author Response · Authors · 2025-08-05
> > > **Thank You and Respectful Request for Reconsideration**
> > >
> > > Dear Reviewer qYip,
> > >
> > > Thank you very much for taking the time to carefully review our rebuttal and for your positive response. We greatly appreciate your thorough engagement with our work and your willingness to consider our clarifications regarding the technical contributions, theoretical analysis, and experimental evaluation.
> > >
> > > Your feedback has been invaluable in helping us better articulate the advantages and practical implications of our approach. We are pleased that our detailed responses have addressed your concerns satisfactorily.
> > >
> > > Since we have clarified that the misunderstandings regarding our theoretical analysis (which is actually independent of SWEET) and other technical aspects do not actually exist in our current submission, we would be deeply honored if you might consider giving our work a further evaluation in light of these clarifications.
> > >
> > > Thank you once again for your constructive review process and professional approach to the discussion.
> > >
> > > Best regards,
> > >
> > > The authors of "Practical and Effective Code Watermarking for Large Language Models"

---

### Official Review · Reviewer_LCcD · 2025-06-30

**Clarity:** 3
**Significance:** 2
**Originality:** 3
**Rating:** 4
**Confidence:** 3

**Summary:**

This paper proposes ACW to address the trade-off between detectability and quality in code watermarking. As a low-entropy text genre, code offers limited positions for watermark insertion, creating an inherent tension between detectability and code integrity. Specifically, not every output token in code generation can embed a watermark, and even at modifiable positions, precise manipulation of model logits is required to avoid compromising code functionality. To resolve this challenge, the authors present an AST-based approach for generating and identifying watermarks in code. The method leverages AST to generate alternative code segments, enabling entropy calculation for each token that reflects its generative flexibility - watermark embedding only proceeds when this entropy exceeds a predefined threshold. Subsequently, ACW also generates logits for each token, sorts them in descending order, groups adjacent tokens as semantic equivalents, and randomly enhances one from each pair via random selection to embed the watermark. Experimental results demonstrate the method's effectiveness in code generation and its cross-linguistic generalizability.

**Questions:**

Can you justify why AST-assisted training would provide better entropy and logits than origianl model ones?
Also, provide more insights about why the performance is sometimes poorer than baselines.
How many alternative code segment would AST create in general?

**Ethical Concerns:**

["NO or VERY MINOR ethics concerns only"]

**Final Justification:**

One major concern of mine is that why not use the original model entropy and logits during the watermarking process, which is well justified by the authors as model may be inaccessible sometimes due to various conerns.

**Limitations:**

yes

**Quality:**

3

**Strengths And Weaknesses:**

Strengths
1. Using AST to acquire alternative code segments is reliable and efficient.
2. The experiments also include datasets of different programming languages.

Weaknesses:
1. ACW produces token entropy and logits based on AST-assisted training of a code dataset. It is not justified why entropy and logits from the model itself fail to achieve the same effect as those produced by ACT as they are pretty much the same concept. The dataset for training is not well explaint with detailed stats. Since the pretrained data of a code LLM is much much larger than a single dataset used here, it is reasonable to believe that the original model entropy and logits are better.
2. The performance on pass@k for ACW is sometimes inferior to other baselines, especially for JS, and no explanation is provided.

---

> ### Author Rebuttal · Authors · 2025-07-31
>
> Dear reviewer LCcD,
>
> We sincerely appreciate your detailed comments and thoughtful evaluation. We believe there may be some misunderstandings regarding our approach, so we would like to provide a comprehensive point-by-point response to clarify our contributions.
>
> ------
>
> ## 1. AST-based vs. Original Model Entropy/Logits
>
> **Reviewer concern: "It is not justified why entropy and logits from the model itself fail to achieve the same effect as those produced by ACW... the original model entropy and logits are better."**
>
> **Response:** We thank the reviewer for this important question. Our AST-based approach is primarily motivated by a critical real-world constraint: **the detector cannot access the original LLM** during deployment.
>
> As stated in our threat model (Line 82), our system is designed for scenarios involving proprietary or API-based models, where obtaining the original model's logits or entropy at detection time is impossible. Our method is therefore not an alternative to using model logits, but rather a necessary solution for when they are unavailable—a common scenario in practical deployments.
>
> However, even when LLM access is available, our solution demonstrates superior performance due to several fundamental advantages. Actually, if we use original LLM information to help the detection, you can think of it as some variant of SWEET.
>
> [1] Who Wrote this Code? Watermarking for Code Generation, ACL 2024.
>
> ### a) Distribution Shift and Context Mismatch
>
> LLMs and watermarking operate in fundamentally different contexts:
>
> - **Training Context**: LLMs are trained on complete texts (thousands of tokens) with full semantic context
> - **Watermarking Context**: Watermarking decisions must be made with minimal context (2-10 tokens) and often apply to code fragments rather than complete programs
>
> This mismatch creates systematic calibration issues. To demonstrate this, we conducted additional experiments using SWEET variants under conditions that mirror our method's constraints:
>
> **Table 1: Fair Comparison with SWEET Variants**
>
> | Dataset   | Method                 | Pass@1 (%) | AUROC (%) | TPR (%)   |
> | --------- | ---------------------- | ---------- | --------- | --------- |
> | HumanEval | SWEET (No Prompt)      | 60.46      | 76.24     | 27.44     |
> |           | SWEET (Context Window) | 61.65      | 71.19     | 17.07     |
> |           | **ACW (Ours)**         | **64.02**  | **84.43** | **45.12** |
> | MBPP      | SWEET (No Prompt)      | 39.64      | 77.24     | 24.80     |
> |           | SWEET (Context Window) | 42.06      | 76.57     | 28.40     |
> |           | **ACW (Ours)**         | **41.32**  | **81.18** | **44.20** |
>
> The results clearly show that SWEET's performance degrades significantly under realistic deployment conditions (no prompt access and limited context), while ACW maintains superior performance across all metrics.
>
> ### b) Computational Efficiency
>
> Our approach offers substantial efficiency advantages:
>
> - **Memory Usage**: Our WPN requires <0.5GB regardless of precision, representing <1/10 of a 1.5B model's memory and becoming negligible for larger models
> - **Computational Overhead**: Watermarking computation can execute in parallel with LLM inference, with actual delay <100μs for tensor slicing operations
> - **Scalability**: The watermark model's overhead becomes increasingly negligible as base model size grows
>
> **Table 2: Computational Efficiency Comparison**
>
> | Model                 | Parameters | Memory Usage | Actual Delay                |
> | --------------------- | ---------- | ------------ | --------------------------- |
> | **ACW (Ours)** | 118M       | < 0.5GB      | < 100 μs (tensor slicing)   |
> | Baseline 1.5B Model   | 1,500M     | ~3-6GB       | ~500-800ms (per-token, GPU) |
>
> This demonstrates that our approach achieves substantial performance improvements while maintaining negligible computational overhead, making it highly practical for production deployment.
>
> ------
>
> ## 2. Pass@k Performance and JavaScript Results
>
> **Reviewer concern: "Performance on pass@k for ACW is sometimes inferior to other baselines, especially for JS."**
>
> **Response:** Thank you for this important observation. We provide clarification on two key points:
>
> ### a) EXP-edit Performance Analysis
>
> The reviewer correctly notes performance differences with EXP-edit. However, EXP-edit's results exhibit concerning instability, as also observed in SWEET [1] (page 17, Appendix E.3). This instability stems from EXP-edit being a sampling-based method that differs fundamentally from green-red watermarking approaches.
>
> **Table 3: EXP-edit Stability Analysis**
>
> | Dataset   | Method                  | Pass@1 (%) | Pass@10 (%) | AUROC (%) | TPR (%)   |
> | --------- | ----------------------- | ---------- | ----------- | --------- | --------- |
> | HumanEval | EXP-edit (original run) | 49.09      | **83.06**   | 66.50     | 25.61     |
> |           | EXP-edit (second run)   | 59.29      | 72.41       | 45.17     | 4.27      |
> |           | **ACW (Ours)**          | **64.02**  | 79.22       | **84.43** | **45.12** |
>
> Our method provides more stable and reliable performance across both functionality and detectability metrics.
>
> ### b) JavaScript-Specific Analysis
>
> For JavaScript, ACW's Pass@1 score (56.33%) is marginally below EXP-edit's (56.59%) by only 0.26%. However, ACW provides a significantly superior trade-off:
>
> - **Detection Performance**: ACW achieves AUROC of 76.94% and TPR of 22.56%, compared to EXP-edit's poor AUROC of 57.68% and TPR of 6.71%
> - **Practical Utility**: The minimal functionality difference is far outweighed by the substantial improvement in watermark reliability
>
> We hypothesize that JavaScript's syntactic flexibility presents unique challenges for watermarking due to its diverse idiomatic patterns. Nevertheless, ACW consistently demonstrates superior balance between code functionality and watermark detectability across all tested languages.
>
> ------
>
> ## 3. Training Dataset Details
>
> **Reviewer concern: "The dataset for training is not well explained with detailed stats."**
>
> **Response:** We appreciate this feedback and provide additional details beyond those in Appendices E.1 and B:
>
> ### Dataset Composition
>
> - **Source**: Generated solutions from HumanEval and MBPP questions (not ground-truth solutions)
> - **Augmentation**: Each solution generates 10-20 AST variants through our transformations
> - **Scale**: Total training samples of ~15K-20K unique code segments from the code variants
> - **Coverage**: Diverse programming patterns including loops, conditionals, functions, and classes
>
> ### Cross-Domain Evaluation
>
> We also evaluated our method's generalizability using different training sources:
>
> **Table 4: Cross-Domain Training Results**
>
> | Training Dataset                           | Evaluation Dataset | Pass@1 (%) | AUROC (%) | TPR (%) |
> | ------------------------------------------ | ------------------ | ---------- | --------- | ------- |
> | open-r1/verifiable-coding-problems-python  | HumanEval          | 63.57      | 82.51     | 42.68   |
>
>
> This demonstrates that our training approach generalizes well and that domain-specific training provides optimal performance.
>
> ------
>
> ## Conclusion
>
> Our responses demonstrate that:
>
> 1. **Practical Necessity**: Our AST-based approach addresses fundamental deployment constraints that make LLM-dependent methods impractical
> 2. **Superior Performance**: Even under fair comparison conditions, ACW outperforms existing methods in both functionality and detectability
> 3. **Efficiency**: Our approach provides substantial computational advantages while maintaining effectiveness
> 4. **Reliability**: Our method offers more stable and consistent performance compared to alternatives
>
> We believe these clarifications demonstrate that ACW represents a significant advancement in practical code watermarking, providing an effective solution for real-world deployment scenarios where existing methods fail due to access constraints.

---

> ### Author Response · Authors · 2025-08-05
> **Dear Reviewer LCcD: we understand that you maybe busy, so we would greatly appreciate it if you could check out our rebuttal.**
>
> Dear NeurIPS Reviewer LCcD:
>
> We are deeply grateful for the time and effort you have invested in reviewing our paper and for providing such thoughtful and constructive comments regarding our AST-guided code watermarking approach. Your insights have been invaluable in helping us improve our work.
>
> We have carefully considered each of your concerns and have done our best to address them comprehensively in our detailed rebuttal. Specifically, we have provided additional explanations regarding the justification for AST-based entropy over original model entropy/logits, offered further analysis of the performance results including the JavaScript case, and included the training dataset details you kindly requested.
>
> We recognize that your questions touched on important aspects of our method's contributions and experimental evaluation, and we hope our point-by-point responses with additional experimental evidence have been helpful in clarifying these matters.
> We understand that you may be very busy, but as the discussion period draws to a close, we would be extremely grateful if you could take a moment to review our rebuttal when convenient. If there are any remaining questions or if any aspects require further clarification, we would be more than happy to provide additional information.
>
> We sincerely hope that our responses have addressed your concerns, and we would be honored if you might consider our clarifications in your final assessment.
> Thank you once again for your expertise, patience, and dedication to the review process. Your feedback has been tremendously valuable to us.
>
> With sincere appreciation,
>
> The authors of "Practical and Effective Code Watermarking for Large Language Models"

---

### Official Review · Reviewer_9ELY · 2025-07-03

**Clarity:** 4
**Significance:** 4
**Originality:** 4
**Rating:** 6
**Confidence:** 4

**Summary:**

This paper presents ACW, a watermarking technique for code generation. ACW relies on the insight that traditional LLM watermarks do not work well for coding applications because code is more sensitive to token changes than regular text. Instead, ACW uses an auxiliary network to determine which tokens can be watermarked without breaking the code’s functionality, and it leverages the existing green/red watermarking technique introduced by  [16]. Through extensive evaluation, the paper demonstrates that ACW is a state‑of‑the‑art practical watermark (one that can be detected without needing the original model as an oracle) and that it has a minimal negative impact on code quality.

**Questions:**

Thank you for submitting your paper to NeurIPS! I really enjoyed reading your work, and I think it is a strong contribution to the watermarking space. I have one important clarification question; the remainder of my comments are guided by curiosity and are not strictly necessary to include in the paper.

Main clarification: Is the WPN trained once and then used for all generations, or is it trained anew for every generation? I assume it is trained once and reused, but specifying this—as well as providing details about the training data—would be helpful for the reader. One possible concern is that if the WPN is trained on the same data used for generation, the technique might not generalize well to unseen coding tasks.

Other comments:
- The WPN is trained on prefixes that are repeated in variants of the same coding sample. This means that, at training time, the model will only see prefixes that cannot be watermarked; if they could be, the AST‑driven variants would change those prefixes to different but equivalent forms. I wonder if this might affect performance and whether introducing more diverse data could counteract this.
- I find it interesting that the delta in Pass@1 for the larger model is smaller than for the other models. My intuition is that smaller models are less proficient at coding tasks, and their output probabilities for code tokens that cannot be changed (such as "def", or reusing an existing variable name) have too much entropy, meaning watermarks that do not use your technique can accidentally break the code. In contrast, larger models are better at coding, and the next‑token probabilities for such tokens have very low entropy, so even standard watermarking techniques are unlikely to choose the wrong token.

**Ethical Concerns:**

["NO or VERY MINOR ethics concerns only"]

**Final Justification:**

I stand by my review, i think this is great, practical work, addressing a task that most watermarking schemes fail at.

**Limitations:**

yes

**Paper Formatting Concerns:**

No concerns

**Quality:**

4

**Strengths And Weaknesses:**

## Strengths
- The paper is very well written and easy to read.
- The underlying idea is simple yet effective. This design paradigm could be applied to other structured tasks where watermarking is challenging and domain knowledge about the expected structure can guide the process.
- Performance is good, and, most importantly, code quality is largely preserved.
- The evaluation is detailed and extensive. Multiple parameter choices are tested, and the method is compared against several baselines. I appreciate the inclusion of experiments across different programming languages.

## Weaknesses
- More details about the training data used for the WPN would be helpful.
- A runtime performance analysis could be valuable.
- I did not see at which FPR threshold the TPR was computed. In general, metrics such as TPR at a fixed FPR would help convey how effective the watermarks are in practical settings. I would also like to see an evaluation of how many tokens are needed to detect the watermark at a fixed FPR.

---

> ### Author Rebuttal · Authors · 2025-07-31
>
> Dear Reviewer 9ELY,
>
> We sincerely appreciate your thorough review and positive evaluation of our work. Your insightful questions help us clarify important aspects of our approach. We address each of your points below:
>
> ------
>
> ## Main Clarification: WPN Training and Generalization
>
> Thank you for this excellent question—it touches on a core aspect of our method. **The WPN is trained once and then reused for generation**, but the relationship between training data and performance is nuanced and worth explaining in detail.
>
> Let me share some key insights about this problem. The starting point for understanding our approach is quite simple: our baseline WLLM performs random partitioning and places watermarks everywhere without any intelligence. Thus, any improvement over this baseline is relatively straightforward to achieve.
>
> Our method improves upon this baseline in two key ways:
>
> 1. **Good Selection**: We need some correlation with code structure. While LLM entropy works, our branching entropy is better and, crucially, doesn't require LLM access during detection.
> 2. **Good Partition**: We need some correlation with good token choices, rather than random partitioning.
>
> In fact, **if you deploy a WPN with no training whatsoever, you essentially get WLLM performance**—this demonstrates that any relevant training (learn code structure or token distribution) will bring improvement over the random baseline. This emphasizes how straightforward it is to surpass existing methods once you move beyond completely random approaches.
>
> ### Cross-Domain Generalization Results
>
> Based on this understanding, our method demonstrates strong generalization capabilities:
>
> - **Cross-dataset**: Training on Python dataset 1, evaluating on dataset 2 → improvement
> - **Cross-language**: Training on Python, evaluating on other languages → improvement (in most experiments)
> - **Multi-language**: Training on mixed-language dataset, evaluating on any language → improvement
>
> The key principle is: **the more closely aligned the training code and evaluation task, the better the performance**.
>
> **Cross-Domain Training Results**
>
> | Training Dataset                           | Evaluation Dataset | Pass@1 (%) | AUROC (%) | TPR (%) |
> | ------------------------------------------ | ------------------ | ---------- | --------- | ------- |
> | open-r1/verifiable-coding-problems-python  | HumanEval          | 63.57      | 82.51     | 42.68   |
> | **HumanEval Generated Solutions Variants** | HumanEval          | 64.02      | 84.43     | 45.12   |
>
> For our main results, the WPN was trained on variants of generated solutions from HumanEval and MBPP questions. We generated solutions using questions from these datasets, created 10-20 AST variants per solution, segmented them, computed branching entropy, and trained the WPN accordingly.
>
> **Important clarification**: While the best results in our tables are achieved when training and evaluation domains are well-aligned, **a general WPN model still achieves good performance**.
>
> ------
>
> ## Runtime Performance Analysis
>
> **Computational Efficiency Comparison**
>
> | Model               | Parameters | Memory Usage | Actual Delay                |
> | ------------------- | ---------- | ------------ | --------------------------- |
> | **ACW (Ours)**      | 118M       | < 0.5GB      | < 100 μs (tensor slicing)   |
> | Baseline 1.5B Model | 1,500M     | ~3-6GB       | ~500-800ms (per-token, GPU) |
>
> **Key advantages:**
>
> - **Memory Usage**: WPN requires <0.5GB regardless of precision, representing <1/10 of a 1.5B model's memory
> - **Computational Overhead**: Watermarking computation executes in parallel with LLM inference, with the primary delay being tensor slicing operations
> - **Scalability**: Overhead becomes increasingly negligible as base model size grows
>
> This demonstrates that our approach achieves substantial performance improvements while maintaining negligible computational overhead for production deployment. The WPN's 118M parameters are minimal compared to modern LLMs, and the parallel execution design ensures real-time feasibility.
>
> ------
>
> ## Token Requirements for Watermark Detection
>
> **Response to your question about how many tokens are needed to detect the watermark at a fixed FPR:**
>
> We provide a detailed analysis of watermark density and detection requirements across our evaluation datasets:
>
> **Watermark Detection Token Analysis**
>
> | Dataset   | Avg Sequence Length | Watermark Fraction (%) | Avg Watermarked Tokens |
> | --------- | ------------------- | ---------------------- | ---------------------- |
> | HumanEval | 71                  | 28.60%                 | ~20.3                  |
> | MBPP      | 39                  | 39.53%                 | ~15.4                  |
>
> This analysis reveals several important insights:
>
> 1. **Selective Watermarking**: Our method only watermarks ~28-40% of tokens, focusing on positions where safe modifications are possible without breaking functionality
> 2. **Sufficient Detection Material**: Even with selective watermarking, we achieve 15-20 watermarked tokens per code snippet on average, providing adequate statistical power for detection at 5% FPR
> 3. **Quality-Detection Balance**: Higher watermark fractions (MBPP: 39.53%) indicate more watermarking opportunities in shorter, simpler code segments
>
> The statistical detection framework requires sufficient watermarked tokens to achieve reliable detection at 5% FPR. Our analysis shows that typical code snippets provide adequate watermarked content for robust detection while maintaining code functionality.
>
> ------
>
> ## Evaluation Metrics
>
> Regarding the FPR threshold for TPR computation: As mentioned in line 277, we use **TPR@5%FPR** for our evaluations. This provides a practical assessment of watermark effectiveness in real-world settings where maintaining low false positive rates is crucial.
>
> ------
>
> ## Observation on Larger Models
>
> Your observation about the smaller delta in Pass@1 for larger models is fascinating and reveals an interesting aspect of our approach. This result actually represents a **transfer experiment**—the WPN was trained on the 1.5B model but applied to the 8B model without retraining.
>
> This demonstrates another dimension of WPN's generalization capability: **cross-model transfer**. The 8B model performs even better than the 1.5B model with the same WPN, suggesting that larger models can better leverage the watermarking guidance provided by our trained network.
>
> However, cross-model generalization does have limitations—we require the same tokenizer since partitioning depends on consistent vocabulary and tokenization. Your intuition about larger models having better-calibrated probabilities for critical code tokens is insightful and likely contributes to this improved transfer performance.
>
> ------
>
> ## Additional Comments
>
> Regarding your observation about training on repeated prefixes: This is an astute point. While our AST-driven variants do change some prefixes, many structural elements remain consistent, providing the model with diverse examples of both watermarkable and non-watermarkable positions. The diversity comes from the interaction between different code structures and the varying contexts in which similar constructs appear.
>
> ------
>
> Thank you again for your thoughtful review and constructive feedback. We believe these clarifications demonstrate both the practical utility and theoretical soundness of our approach.

---

> > ### Comment · Reviewer_9ELY · 2025-08-05
> >
> > Thank you for your detailed response. I missed the TPR@5%FPR mention on line 277, thanks for pointing that out. One nit is that 5% FPR is actually quite high for watermarking, maybe reporting a second number at a lower rate would be interesting as well. I appreciate all these additional experiments.

---

> > > ### Author Response · Authors · 2025-08-05
> > >
> > > Thank you for this excellent suggestion! You're absolutely right that 5% FPR is quite high for watermarking applications. We will conduct additional experiments at lower FPR thresholds (e.g., 1%) to provide a more comprehensive evaluation in our final version.
> > >
> > > We initially adopted the 5% FPR setting from [1] to maintain consistency with prior work, but your point about practical deployment requirements is well-taken. These additional metrics will better demonstrate our method's performance in more stringent real-world scenarios.
> > >
> > > Thank you again for your thoughtful feedback. If you have any other experiments or suggestions you'd like to see in future version, please feel free to add them here. We'll do our best to address them and refine our work accordingly!
> > >
> > > [1] Who Wrote this Code? Watermarking for Code Generation, ACL 2024.

---

### Official Review · Reviewer_39QH · 2025-07-03

**Clarity:** 3
**Significance:** 3
**Originality:** 3
**Rating:** 4
**Confidence:** 4

**Summary:**

This paper proposes ACW (AST-guided Code Watermarking), a novel watermarking framework for LLM-generated code. ACW combines Abstract Syntax Tree (AST) analysis with a learnable Watermark Partitioning Network (WPN) to identify semantically substitutable positions for watermark insertion.

**Questions:**

Questions and Suggestions
1.What is the inference latency introduced by the WPN, especially given its transformer-based design? Please quantify its runtime cost and potential bottlenecks.
2.How should δ, γ, and α be tuned in real-world applications? Can the authors offer practical guidance on balancing detection performance with code fidelity?

**Ethical Concerns:**

["NO or VERY MINOR ethics concerns only"]

**Final Justification:**

Maintain the original score.

**Limitations:**

While some limitations are briefly discussed (e.g., language generalization, WPN training), several practical concerns remain unaddressed:
1. The need to independently implement AST parsers and transformations for each target language adds substantial engineering overhead and may limit the method’s scalability.
2. It is unclear how detection consistency is ensured across different WPN versions or deployments — i.e., what happens when model drift occurs.
3. The authors do not discuss how resilient the method is to adversaries attempting to reverse-engineer green-token distributions.

**Paper Formatting Concerns:**

N

**Quality:**

3

**Strengths And Weaknesses:**

Strengths
Black-box detection: No LLM weight/prompt access required.
Code preservation: AST-guided embedding maintains functionality.
Attack resistance: Robust to renaming and paraphrasing (DIPPER).

Weaknesses
 Lack of justification for excluding recent code watermarking method such as CodeIP
CodeIP: A Grammar-Guided Multi-Bit Watermark for Large Language Models of Code. Findings of ACL: EMNLP 2024
 Potentially underestimated training and inference overhead: WPN is a 6-layer Transformer with a 512-dimensional hidden size and must run a forward pass at every generation step. During training, n-gram grouping and entropy computation over AST-transformed code variants are required. However, the paper does not report latency or GPU usage, which could significantly increase inference cost compared to non-watermarked generation and impact real-time deployment feasibility.
 Dependence on high-quality AST transformation data: ACW relies on functionally equivalent code variants labeled by branching entropy for supervision. This necessitates accurate AST parsing and rule-based code transformations per language. In environments lacking mature parsers (or with complex macros/templates), generating high-coverage training data may be infeasible, reducing portability.
 Lack of detailed parameter sensitivity analysis: The paper uses fixed hyperparameters (δ = 2, γ = 0.5) and briefly discusses an inverse-U curve with respect to α (switch threshold). However, it does not explain how to choose optimal parameters under different code lengths, risk tolerances, or application settings. It also omits analysis of trade-offs between detection strength and potential degradation in code quality or watermark detectability.

---

> ### Author Rebuttal · Authors · 2025-07-31
>
> Dear Reviewer 39QH,
>
> We appreciate your thorough review and constructive feedback. Your concerns about practical deployment considerations are important, and we're pleased to address each point with detailed explanations and experimental evidence.
>
> ------
>
> ## 1. Inference Latency and Computational Overhead
>
> **Response to your concern about WPN's transformer-based design and runtime costs:**
>
> Although we introduce the WPN, the overhead is minimal even for large-scale deployments. Our watermark model requires **less than 0.5GB** of memory regardless of precision (full or half), representing a tiny fraction of the main model's memory footprint. For instance, this accounts for **less than 1/10 of a 1.5B model's memory and becomes even more negligible for larger models (7B, 64B)**. The context window requirement (less than 10 tokens) further limits memory usage. Most importantly, the watermarking computation **can be executed in parallel with LLM inference**, meaning the only additional costs are minimal memory usage and the delay caused by slicing operations on the vocabulary, which typically takes **less than 100 μs**.
>
> **Computational Efficiency Comparison**
>
> | Model    | Parameters | Memory Usage | Actual Delay                |
> | -------- | ---------- | ------------ | --------------------------- |
> | WPN      | 118M       | < 0.5GB      | < 100 μs (tensor slicing)   |
> | 1.5B LLM | 1,500M     | ~3-6GB       | ~500-800ms (per-token, GPU) |
>
> This demonstrates that our approach achieves substantial performance improvements while maintaining negligible computational overhead for production deployment. The WPN's 118M parameters are minimal compared to modern LLMs, and the parallel execution design ensures real-time feasibility.
>
> **Regarding training overhead**: While training requires n-gram grouping and entropy computation over AST-transformed code variants, training is performed only once. Therefore, training complexity does not affect inference delay in production deployments.
>
> ------
>
> ## 2. Cross-Language Generalization and AST Parser Dependencies
>
> **Response to concerns about AST parsing complexity and language portability:**
>
> Most importantly, our approach learns generalizable rules about code syntax and structure that transfer across different programming languages. To demonstrate this crucial capability, we conducted **experiments on the HumanEvalPack dataset**, testing our model (trained exclusively on Python) on Java and C++ without any additional fine-tuning.
>
> **Cross-Language Generalization Results**
>
> | Language | Method         | Pass@1 (%) | AUROC (%) | TPR (%)   |
> | -------- | -------------- | ---------- | --------- | --------- |
> | Java     | WLLM           | 24.02      | 42.27     | 6.10      |
> |          | **ACW (Ours)** | **26.72**  | **65.31** | **24.25** |
> | C++      | WLLM           | 44.27      | 65.13     | 21.19     |
> |          | **ACW (Ours)** | **55.33**  | **78.54** | **24.62** |
>
> These results clearly demonstrate that **ACW generalizes effectively across programming languages**, significantly outperforming practical baselines in most cases.
>
> **Regarding AST parser implementation**: Every programming language, by necessity, must have parsers to achieve compilation or interpretation—it's a fundamental requirement for any language. Moreover, as our cross-language results show, programming languages share sufficient structural similarities that transfer learning is highly effective. The strong performance can be understood intuitively: existing non-model-based approaches rely primarily on random partitioning, so even a moderately intelligent model can substantially surpass such methods.
>
> Additionally, we believe that pretraining on large datasets with multiple programming languages can further address scalability concerns and reduce the need for language-specific implementations.
>
> ------
>
> ## 3. Parameter Sensitivity Analysis
>
> **Response to concerns about hyperparameter tuning guidance:**
>
> We provide comprehensive analysis of key parameters. In our experiments, we intentionally fixed the hyperparameters as we consider them watermarking budget parameters. We fixed them to ensure fair comparisons across different green-red watermarking methods.
>
> **Delta (δ) Analysis** - As shown in Table 4 of our paper, we demonstrate the trade-off between watermark strength and code quality.
>
> **Gamma (γ) Analysis** - We conducted experiments showing the impact of different green list ratios on the detection-quality trade-off:
>
> | Gamma   | Method | Pass@1 (%) | AUROC (%) | TPR (%) |
> | ------- | ------ | ---------- | --------- | ------- |
> | γ = 0.5 | ACW    | 64.02      | 84.43     | 45.12   |
> | γ = 0.7 | ACW    | 64.18      | 78.62     | 35.25   |
>
> **Practical Guidance**: These parameters function as watermarking budget controls:
>
> - **δ (bias strength)**: Higher values increase detection accuracy but may reduce code quality
> - **γ (green ratio)**: Affects the statistical balance for detection
> - **α (switch threshold)**: Controls selective watermarking frequency, with higher values reducing watermark insertion but improving code quality
>
> **Regarding trade-offs between detection strength and code quality**: While we cannot include figures in this rebuttal, practitioners can refer to Figure 3 of SWEET [1] as a model for plotting different hyperparameter setups to find suitable trade-offs between Pass@k and AUROC metrics.
>
> The beauty of our approach is that **any training in the correct direction improves performance** over the random baseline, making parameter optimization less critical than in other methods.
>
> [1] Who Wrote this Code? Watermarking for Code Generation, ACL 2024.
>
> ------
>
> ## 4. Detection Consistency Across Deployments
>
> **Response to concerns about model drift and deployment consistency:**
>
> Once the WPN is trained, it functions as a **fixed deterministic function** paired with a security key. The detection process is entirely deterministic:
>
> 1. **Fixed WPN**: The trained model parameters remain constant across deployments
> 2. **Security Key**: Ensures consistent green-red partitioning using cryptographic functions
> 3. **Deterministic Process**: Same input always produces same watermarking decisions
>
> This design ensures perfect consistency across different deployments and eliminates concerns about model drift during detection.
>
> ------
>
> ## 5. Security Against Reverse Engineering
>
> **Response to adversarial resilience concerns:**
>
> As detailed in **Appendix G**, our method provides robust protection against reverse engineering attempts:
>
> 1. **Model Complexity**: The WPN's neural network architecture with high-dimensional representations makes model extraction computationally challenging
> 2. **Cryptographic Security**: Integration with Pseudorandom Functions (PRF) and security keys makes watermark pattern prediction computationally intensive
> 3. **Dual Protection**: Adversaries would need to both reverse-engineer the complex model decision process AND obtain the secret key
>
> This multi-layered approach significantly raises the bar for potential attacks compared to traditional hash-based or fixed-key methods.
>
> ---
>
> ## Response to CodeIP Comparison
>
> **Regarding the exclusion of CodeIP comparison:**
>
> We acknowledge CodeIP as an important recent work in code watermarking. However, we encountered several fundamental compatibility issues that prevented a fair comparison:
>
> 1. **Dataset Compatibility**: In CodeIP Section 4.1, the authors explicitly mention that their model is not suitable for HumanEval and MBPP benchmarks, which are our primary evaluation datasets.
> 2. **Context Window Limitations**: CodeIP's approach requires extensive context and cannot operate effectively within the limited context windows that our method is designed to handle.
> 3. **Fundamental Setting Differences**:
>    - **Multi-bit vs. Binary**: CodeIP targets multi-bit watermarking while our approach focuses on binary detection
>    - **Prompt Dependency**: As described in CodeIP Section 3.2 (Lexical Token Type Predictor), their predictor requires access to the original prompt during detection, which contradicts our practical deployment setting where prompts are unavailable
>    - **Oracle Requirements**: CodeIP's design assumes access to generation context that our threat model explicitly excludes
> 4. **Implementation Challenges**: Despite our efforts to implement CodeIP for comparison, the combination of different assumptions, requirements, and evaluation settings made it technically infeasible to establish a fair experimental comparison within our framework.
>
> We attempted to adapt CodeIP to our experimental setting but found that the fundamental architectural and assumption differences prevented meaningful comparison. We plan to explore this comparison in future work with modified experimental protocols, but completing this analysis within the rebuttal timeframe proved technically challenging.
>
> ------
>
> ## Additional Comments
>
> **Baseline Simplicity**: A key insight is that our baseline (WLLM) uses completely random partitioning. In fact, **deploying an untrained WPN essentially yields WLLM performance**, demonstrating that any relevant training provides improvement. This fundamental simplicity of existing methods makes our improvements both significant and achievable.
>
> ------
>
> We believe these clarifications address your practical deployment concerns and demonstrate that ACW provides a robust, efficient, and scalable solution for real-world code watermarking applications.
>
> Thank you for your thoughtful review and valuable feedback.

---

> > ### Comment · Reviewer_39QH · 2025-08-05
> > **Thanks for the clarification. No further concerns.**
> >
> > Thanks for the clarification. No further concerns.

---

### Note · Authors · 2025-08-12

Dear AC and Reviewers:

Thank you for your time and effort in reviewing our paper. We would like to briefly summarize our rebuttal efforts and key points:

- We have provided comprehensive responses to all reviewer concerns with extensive experimental evidence. Key results include: (1) **Cross-model generalization** - our WPN trained on a 1.5B model successfully transfers to an 8B model without retraining, achieving 81.22% AUROC (WLLM 65.90% AUROC); (2) **Cross-language generalization** - training only on Python, ACW achieves 78.54% AUROC on C++ (WLLM 65.13% AUROC); (3) **Minimal computational overhead** - requiring only <0.5GB memory and <100μs delay through parallel execution with LLM inference; (4) **Robustness** - demonstrated resilience against GPT-4 refactoring attacks. We have also **provided implementation code** for reference. We are pleased that Reviewers 9ELY and 39QH found our clarifications satisfactory and engaged constructively with our work.

- We invested significant effort in addressing Reviewer LCcD's concerns about LLM as WPN justification (Note: Appendix C.3 already discusses limitations of using LLMs as WPN), providing detailed responses with additional experimental evidence. While we did not receive feedback during the rebuttal period, we sincerely appreciate their comments. We will carefully consider their feedback and incorporate improvements in our final version.

- We appreciate Reviewer qYip's thoughtful questions. We recognize there were some initial misunderstandings that we were able to clarify.  We provided clarifications on several key points, including that our theoretical analysis is completely independent of SWEET, and that our WPN demonstrates successful transfer learning across model sizes without requiring retraining. We are grateful that the reviewer acknowledged our clarifications with no further concerns, and we believe we have thoroughly addressed all points raised.

**In summary:** As LLM-generated code becomes increasingly prevalent in software development, code watermarking is becoming essential for attribution and intellectual property protection. We believe ACW makes meaningful contributions to this important and timely field by providing a practical, efficient solution that works without LLM access during detection—addressing real-world deployment constraints.

Thank you again to all reviewers and the AC for your valuable feedback.

---

### Decision · Program_Chairs · 2025-09-17

**Decision:**

Accept (poster)

**Comment:**

This paper introduces ACW (AST-guided Code Watermarking), which is designed for watermarking LLM-generated code in a black-box setting (i.e. no access to logits). The core idea is to use Abstract Syntax Trees (ASTs) to guide the watermarking process. By training a small model (the Watermark Partitioning Network) on AST-derived features, the system learns to identify positions in the code where a watermark can be embedded without breaking functionality.

The paper's primary strengths are its interesting domain-specific approach for code (which is challenging) and its large amount of empirical validation, which was strengthened during the review. The approach achieves strong detection rates while preserving code functionality.

The initial submission had some unanswered questions, such as its computational overhead and its ability to work across different languages and model sizes. However, the rebuttal addressed these points with new experiments that mostly satisfied reviewers. They demonstrated that the overhead is negligible, that the method works well in other languages.

Overall this is a complete paper and is technically sound with an interesting and effective idea.